# DU-Shapley: A Shapley Value Proxy for Efficient Dataset Valuation

**Felipe Garrido-Lucero\***
Inria, Fairplay joint team
Palaiseau, France
`felipe.garrido-lucero@irit.fr`
\* Equal contribution

**Benjamin Heymann\***
Criteo AI Lab
Paris, France
`b.heymann@criteo.com`
\* Equal contribution

**Maxime Vono\***
Criteo AI Lab
Paris, France
`m.vono@criteo.com`
\* Equal contribution

**Patrick Loiseau**
Inria, Fairplay joint team
Palaiseau, France
`patrick.loiseau@inria.fr`

**Vianney Perchet**
ENSAE, FairPlay joint team
Palaiseau, France
`vianney@ensae.fr`

## Abstract

We consider the *dataset valuation problem*, that is, the problem of quantifying the incremental gain, to some relevant pre-defined utility of a machine learning task, of aggregating an individual dataset to others. The Shapley value is a natural tool to perform dataset valuation due to its formal axiomatic justification, which can be combined with Monte Carlo integration to overcome the computational tractability challenges. Such generic approximation methods, however, remain expensive in some cases. In this paper, we exploit the knowledge about the structure of the dataset valuation problem to devise more efficient Shapley value estimators. We propose a novel approximation, referred to as discrete uniform Shapley, which is expressed as an expectation under a discrete uniform distribution with support of reasonable size. We justify the relevancy of the proposed framework via asymptotic and non-asymptotic theoretical guarantees and illustrate its benefits via an extensive set of numerical experiments.

## 1 Introduction

One of the main challenges for training machine learning (ML) models with enough generalization capabilities is to access a sufficiently large set of labeled training data. These data often exist but are commonly spread across many parties, impairing their usage in a direct and simple way. Real world examples range from the advertising industry, where different retailers hold sets of observations with either similar or complementary features from consented data about browsing and shopping habits of individual users; to the medical sector where hospitals may improve their diagnostics accuracy by sharing their data. By collaborating with each other and pooling their individual datasets together, these *dataset owners* could learn better ML models for their applications. Naturally, many questions raise from such collaborations. Federated learning [8, 9], for example, addresses the issues related to the practical ways that dataset owners can share their data. We consider a complementary problem to the one in federated learning: measuring the additional value each party would obtain by participating in the joint ML effort. In order to compute or estimate compensating rewards allowing to incentivize parties to share their data, a first stage that is commonly considered in the literature is to perform so-called *dataset valuation* [1, 42, 44].

Motivated by natural properties expected for fair valuation, different solution concepts from cooperative game theory [3] have been considered, the Shapley value [40] being arguably the most broadly

38th Conference on Neural Information Processing Systems (NeurIPS 2024).

studied valuation scheme in ML due to its axiomatic justification. Agarwal et al. [1] designed a data marketplace and used the Shapley value to allocate the data among buyers. Tay et al. [44] considered a cooperative environment where agents can jointly train a generative model, from which synthetic data are drawn and distributed to the parties according to their Shapley values. Sim et al. [42] rewarded parties based on the Shapley value and information gain on model parameters. The critical challenge when using the Shapley value is its well-known computational intractability. To cope with it, [1, 44] considered Monte Carlo (MC) approximations, while [42] worked with a small set of three players. This approximation methods, however, remain expensive whenever computing the marginal contributions involve retraining. Moreover, they are generic and do not use the specific structure of the dataset valuation problem at stake, leaving open the possibility to find more adapted approximations for that problem.

The Shapley value was also used in the related problem of *data valuation*. Data valuation measures the contribution of a single data point within a dataset in the training of a given prediction model. Several solution concepts based on the Shapley value have been proposed for the data valuation problem including `Data Shapley` [10, 16], `DShapley` [11, 24], `Beta Shapley` [23] or `CS-Shapley` [38], together with different MC variants to cope with the computational intractability issue. For the data valuation problem, the structure was exploited to give easier-to-compute solutions in certain cases, in particular for the $k$-nearest neighbor problem [12, 15, 25, 26, 35, 41, 46]. Unlike data valuation, however, dataset valuation aims at quantifying the marginal contribution of a *whole dataset* to a given ML task with respect to (w.r.t.) the datasets brought by other dataset owners. Although data and dataset valuation are related problems, they are different and the techniques developed for data valuation cannot be used for the dataset valuation problem that we study (we further develop this point in Section 2.3).

**Contributions.** We consider the dataset valuation problem. Following the ML literature, we model it as a cooperative game whose value function relates to the considered ML task, and aim at estimating the Shapley value to measure the dataset owners contribution. We propose a new way to address the computational intractability issue of the Shapley value. Instead of relying on generic MC approximation schemes, our approximation method leverages the structure of the dataset valuation problem as well as a convergence result for a key random variable of the problem. Our approximation behaves well in many cases, both theoretically and empirically. More specifically, our main contributions can be summarized as follows:

1. We propose `DU-Shapley` (Definition 3), a novel Shapley value approximation that exponentially reduces the number of utility function valuations required for the computation. This is the first dataset valuation approach leveraging the specific structure of the utility function.
2. Based on three different use-cases, we establish asymptotic and non-asymptotic theoretical guarantees for `DU-Shapley`, showing notably that it converges almost surely to the Shapley value as the number of dataset owners grows.
3. We assess the benefits of the proposed methodology using extensive numerical experiments on both Shapley value approximation and dataset valuation use-cases. We show, in particular, that `DU-Shapley` outperforms all considered MC approximations of the Shapley value.

**Additional Related Work.** Cooperative game theory has been applied to solve multi-agents ML problems beyond data and dataset valuation [6, 18, 29, 47]. In particular, the Shapley value has been used to solve several problems including variable selection [5], feature importance [7, 27, 28], or model interpretation [4]. In these problems, similarly to the data and dataset valuation problems, the computational intractability issue of the Shapley value is usually addressed via MC [2, 31, 32].

## 2 Problem Formulation and Main Concepts Involved

This section presents the dataset valuation problem we aim to solve, along with preliminaries including the definition and classical approximations of the Shapley value. For $n \in \mathbb{N}$ and $A$, we denote $[n] := \{1, .., n\}$ and $\mathrm{U}(A)$ the uniform distribution with support on $A$.

### 2.1 Generic Model

We consider a collaborative ML setting involving a set $\mathcal{I}$ of $I = |\mathcal{I}| \in \mathbb{N}^*$ dataset owners, also referred to as players in the sequel, who are willing to cooperate in order to solve a common ML problem. Each player $i \in \mathcal{I}$ is assumed to possess an individual dataset $\mathrm{D}_i = \{(x_i^{(j)}, y_i^{(j)})\}_{j \in [n_i]}$

where $x_i^{(j)} \in \mathcal{X} \subset \mathbb{R}^d$ stands for a feature vector, $y_i^{(j)} \in \mathcal{Y}$ is a label, $n_i = |\mathrm{D}_i|$ refers to the number of data points in $\mathrm{D}_i$, and samples are drawn independently from a player-dependent distribution $p_i$, i.e., $(x_i^{(j)}, y_i^{(j)}) \sim p_i$, for all $j \in [n_i]$ and $i \in \mathcal{I}$.

Our basic motivation is to quantify the incremental contribution that a given player $i \in \mathcal{I}$ brings by sharing her dataset $\mathrm{D}_i$ with other players towards solving some ML task. Hence, we are interested in scenarios in which, even though the data distribution might differ across players, they face a similar ML task, for instance the minimization of the expectation (with respect to $p_i$) of some loss function $\ell(\hat{Y}, Y)$, where $\hat{Y}$ denotes a prediction of $Y$. In such cases, players can usually learn from others' datasets, in the sense that given some $X$, the optimal prediction $\hat{Y}$ that minimizes $\mathbb{E}[\ell(\hat{Y}, Y)|X]$ is the same for all player. This holds, *e.g.*, if the conditional distributions (or, in many cases, simply the conditional expectation) of $y^{(j)}$ given $x^{(j)}$ are the same but the marginal distributions of $x^{(j)}$ differ.

To model this problem with full generality, we assume that the players $i \in \mathcal{I}$ collaborate in solving an ML task whose success is measured through some abstract metric $u$ that maps any dataset to a real number (say, the prediction accuracy in a classification problem). With a slight abuse of notation, for any coalition of players $\mathcal{S} \subseteq \mathcal{I}$, we define $u(\mathcal{S}) = u(\mathrm{D}_\mathcal{S})$, where $\mathrm{D}_\mathcal{S} := \cup_{i \in \mathcal{S}} \mathrm{D}_i$. Hence, $u : 2^\mathcal{I} \to \mathbb{R}$ can be seen as a game-theoretical utility function that quantifies how well coalitions of players can solve the considered ML task based on the union of their datasets.

The following subsections provide three theoretical use-cases that instantiate the generic model and give specific utility functions $u$ to illustrate the dataset valuation problem. Using different tools and techniques, Section 3 provides theoretical guarantees in each of them. These theoretical results are then complemented in Section 4 by numerical evidence of our proposed approach in more intricate practical problems on real data.

### 2.1.1 Theoretical use-case 1: Non-parametric Regression

The first use case we shall investigate is quite generic and consists in non-parametric regression. We assume the existence of a function $f^*$ such that $y_i^{(j)} = f^*(x_i^{(j)}) + \eta_i^{(j)}$ with $\eta_i^{(j)}$ i.i.d., and a quadratic loss function. Without regularity assumption on $f^*(\cdot)$, learning can be arbitrarily slow; hence it is usually assumed that this mapping is Lipschitz (or at least $\beta$-Hölder [13, 45]).

The standard estimation method of $f^*$ we shall consider is called the *regressogram* or *binning* (also applied in [13] to study local differential privacy within regression) and consists in learning optimal piece-wise constant functions. More precisely, given some parameter $B \in \mathbb{N}$—chosen exogeneously as a function of the function regularity $\beta$, the ambient dimension $d$ and the total number $n$ of datapoints, typically $B \simeq n^{d/(d+2\beta)}$—, the feature space $\mathcal{X}$ is partitioned into $B$ cubic bins. The excess risk of learning $f^*$ can then be decomposed into

$$\mathbb{E}\big[(\hat{f}(x) - f^*(x))^2\big] = \mathbb{E}\big[(\hat{f}(x) - \bar{f}(x))^2\big] + \mathbb{E}\big[(\bar{f}(x) - f^*(x))^2\big], \tag{1}$$

where $\hat{f}$ is the estimator of $f^*$, $\bar{f}(x) := \sum_{b \in [B]} \bar{f}_b \mathbb{1}\{x \in b\}$, and $\bar{f}_b$ is any value that $f^*$ can take on the bin $b$. The second term in (1) being agnostic to the players' datasets, the problem of measuring the contributions of the players to estimating $f^*$ can be decomposed into measuring their contributions to estimating each $\bar{f}_b$. In particular, the utility $u(\mathcal{S})$ of a coalition $\mathcal{S}$ can be defined, and split into the sum of $B$ sub-utilities $u_b(\mathcal{S})$ functions, as follows

$$u(\mathcal{S}) := -\mathbb{E}\big[(\hat{f}_\mathcal{S}(x) - \bar{f}(x))^2\big] = \sum_{b \in [B]} -\mathbb{E}\big[(\hat{f}_{\mathcal{S},b} - \bar{f}_b)^2\big]\mathbb{P}(x \in b) =: \sum_{b \in [B]} u_b(\mathcal{S})\mathbb{P}(x \in b),$$

where $\hat{f}_\mathcal{S}$ is the estimator of $\bar{f}$ when using the datasets of all players in $\mathcal{S}$ and $\hat{f}_{\mathcal{S},b}$ is the estimator $\bar{f}_b$ when using, for all players in $\mathcal{S}$, the datasets of points in the bin $b$. Interestingly, after this reduction, the problem is decomposed into $B$ independent sub-problems—one per bin—, where the utility is a sole function of the number of data points used to estimate $\bar{f}_b$, i.e., we can write $u_b(\mathcal{S}) = w_b(\sum_{i \in \mathcal{S}} n_{i,b})$ for some function $w_b : \mathbb{N} \to \mathbb{R}$, where $n_{i,b}$ is the number of data points that player $i$ has in the bin $b$. This last property motivates our second theoretical use-case.

### 2.1.2 Theoretical use-case 2: Homogeneous case

The second theoretical setting considers a general learning problem (not necessarily restricted to regression) and supposes that all players have the same sampling distribution, i.e., it takes $p_i = p$

for all $i \in \mathcal{I}$. This homogeneity on the players allows to reduce the problem of measuring the contribution of the players to just counting the number of data points contributed by each of them. Formally, and similarly to the previous use-case, we suppose the existence of a function $w : \mathbb{N} \to \mathbb{R}$ such that $u(\mathcal{S}) = w(\sum_{i \in \mathcal{S}} n_i)$.

### 2.1.3   Theoretical use-case 3: Heterogeneous Linear Regression - Local Differential Privacy

The third theoretical setting we consider is linear regression with random design and different variance of the features and labels per player. Although the setting is more general, one of the motivations behind it is standard linear regression with homogeneous data between players, but where players can purposely add noise when sharing their dataset (in order to provide Local Differential Privacy, for instance). Formally, for any $i \in \mathcal{I}$, we consider the following linear model that generates the dataset $\mathrm{D}_i$ of size $n_i$:

$$y_i^{(j)} = x_i^{(j)} \theta + \eta_i^{(j)}, \text{ where } \eta_i^{(j)} \sim \mathrm{N}(0, \varepsilon_i^2), \text{ and } x_i^{(j)} \sim \mathrm{N}(0_d, \sigma_i^2 \mathrm{I}_d), \text{ for any } j \in [n_i], \qquad (2)$$

with $\theta \in \mathbb{R}^d$ a ground-truth parameter, $\sigma_i$ positive and known, and $\varepsilon_i$ the differential privacy level chosen by player $i$. Under the linear regression framework defined in (2), and following [8], the utility function of a set $\mathcal{S} \subseteq \mathcal{I}$ of players is defined by the negative expected mean square error over a hold-out dataset, i.e.,

$$u(\mathcal{S}) = -\mathbb{E}\big[\big(x^\top \hat{\theta}_{\mathcal{S}} - x^\top \theta\big)^2\big], \qquad (3)$$

where the expectation is taken over the distribution $p_{\text{test}}$ of a hold-out testing datum $x \in \mathbb{R}^d$, the sampling distributions $\mathrm{N}(0, \sigma_i^2 \mathrm{I}_d)$ for all $i \in \mathcal{S}$, and the linear regression error distributions $\mathrm{N}(0, \varepsilon_i^2), \forall i \in \mathcal{S}, j \in [n_i]$, and $\hat{\theta}_{\mathcal{S}}$ stands for the generalized least square estimator defined by $\hat{\theta}_{\mathcal{S}} = (X_{\mathcal{S}}^\top \Sigma_{\mathcal{S}}^{-1} X_{\mathcal{S}})^{-1} X_{\mathcal{S}}^\top \Sigma_{\mathcal{S}}^{-1} Y_{\mathcal{S}}$, where $\Sigma_{\mathcal{S}} = \mathrm{diag}((\varepsilon_i^2)_{i \in \mathcal{S}}) \in \mathbb{R}^{|\mathcal{S}| \times |\mathcal{S}|}$. The notations $X_{\mathcal{S}}$ and $Y_{\mathcal{S}}$ refer to the concatenation of $\{X_i\}_{i \in \mathcal{S}}$ and $\{Y_i\}_{i \in \mathcal{S}}$, respectively, and $X_i \in \mathbb{R}^{n_i \times d}$ is defined by $X_i = ([x_i^{(1)}]^\top, \ldots, [x_i^{(n_i)}]^\top)^\top$ while $Y_i \in \mathbb{R}^{n_i}$ is defined by $Y_i = (y_i^{(1)}, \ldots, y_i^{(n_i)})^\top$.

The following result provides a close-form expression for the utility function in this case:

**Proposition 1.** *Let $\mathcal{S}$ be a coalition of players and consider the value function as above. It follows,*

$$u(\mathcal{S}) = \frac{-\mathrm{Tr}\big[\mathbb{E}\big[xx^\top\big]\big]}{q(\mathcal{S}) - d - 1}, \text{ where } q(\mathcal{S}) := \left\lfloor \frac{\big(\sum_{i \in \mathcal{S}} (\sigma_i/\varepsilon_i) n_i\big)^2}{\sum_{i \in \mathcal{S}} \big(\sigma_i/\varepsilon_i\big)^2 n_i} \right\rfloor, \text{ with the convention } q(\varnothing) = 0.$$

*In particular, considering $p_{\text{test}} = \mathrm{N}(0, \mathrm{I}_d)$, we get $u(\mathcal{S}) = \frac{d}{d+1-q(\mathcal{S})}$.*

Proposition 1 shows that, in this use-case, the utility function can be written as a function $w(q(\mathcal{S}))$ of a scalar quantity $q(\mathcal{S})$ that captures the datasets heterogeneity. Notice that in this use-case, if we add the homogeneity assumption that $\sigma_i/\varepsilon_i = \sigma/\varepsilon$, for all $i \in \mathcal{I}$, then the term $q(\mathcal{S})$ becomes $\sum_{i \in \mathcal{S}} n_i$ and, as a consequence, we get

$$u(\mathcal{S}) = w(q(\mathcal{S})) = w\left(\sum_{i \in \mathcal{S}} n_i\right) = \frac{d}{d + 1 - \sum_{i \in \mathcal{S}} n_i}.$$

Recall that, in the non-parametric regression use-case, it holds $u(\mathcal{S}) = \sum_{b \in [B]} \mathbb{P}(x \in b) w_b(q_b(\mathcal{S}))$ where $q_b(\mathcal{S}) = \sum_{i \in \mathcal{S}} n_{i,b}$. Therefore, in our three uses-cases, the utility of a coalition can be summarized as the function of some scalar quantity of interest. This observation will be useful to state later our theoretical results.

## 2.2   Shapley Value

The Shapley value [40] is a classical solution concept in cooperative game theory to fairly allocate the total gains generated by a coalition of players. Given a utility function $u$, the Shapley value of a player $i$ is defined as the average marginal contribution of her dataset $\mathrm{D}_i$ to all possible subsets of $\{\mathrm{D}_j\}_{j \in \mathcal{I} \setminus \{i\}}$, built by aggregating the datasets of the other players. Formally, the Shapley value $\varphi_i$ of player $i$ writes

$$\varphi_i(u) = \frac{1}{|\Pi(\mathcal{I})|} \sum_{\pi \in \Pi(\mathcal{I})} [u(\mathcal{P}_i^\pi \cup \{i\}) - u(\mathcal{P}_i^\pi)], \qquad (4)$$

where $\Pi(\mathcal{I})$ refers to the set of permutations over $\mathcal{I}$ and $\mathcal{P}_i^\pi$ to the set of predecessors of player $i \in \mathcal{I}$ in permutation $\pi \in \Pi(\mathcal{I})$. The Shapley value of player $i$ is equivalently expressed as

$$\varphi_i(u) = \frac{1}{I} \sum_{\mathcal{S} \subseteq \mathcal{I} \setminus \{i\}} \binom{I-1}{|\mathcal{S}|}^{-1} [u(\mathcal{S} \cup \{i\}) - u(\mathcal{S})]. \tag{5}$$

The Shapley value has been commonly used in ML and cooperative game theory as it uniquely satisfies the following set of desirable properties.

1. *Efficiency.* $\sum_{i=1}^{I} \varphi_i(u) = u(\mathcal{I})$, i.e, the sum of all Shapley values is equal to the value of $\mathcal{I}$.
2. *Symmetry.* If, for any $\mathcal{S} \subseteq \mathcal{I} \setminus \{i_1, i_2\}$, $u(\mathcal{S} \cup \{i_1\}) = u(\mathcal{S} \cup \{i_2\})$, then $\varphi_{i_1}(u) = \varphi_{i_2}(u)$, i.e., whenever two players have the same marginal contributions, their Shapley values coincide.
3. *Dummy.* If, for any $\mathcal{S} \subseteq \mathcal{I} \setminus \{i\}$, $u(\mathcal{S} \cup \{i\}) = u(\mathcal{S})$, then $\varphi_i(u) = 0$, i.e., whenever a player has null marginal contributions, her Shapley value is zero.
4. *Linearity.* $\varphi_i(u_1 + u_2) = \varphi_i(u_1) + \varphi_i(u_2)$, i.e., the Shapley value of sums of games is the sum of the Shapley values of the respective games.

**MC approximation of the Shapley Value.** Evaluating the Shapley value is unfortunately computationally expensive in general. As a consequence, many MC approximations have been considered by sampling with replacement $T$ terms from the sum of either (4) or (5). Regarding (4), this boils down to considering the estimator

$$\hat{\varphi}_i(u) = \frac{1}{T} \sum_{t=1}^{T} [u(\mathcal{P}_i^{\pi_t} \cup \{i\}) - u(\mathcal{P}_i^{\pi_t})], \text{ where } \pi_t \sim \mathrm{U}(\Pi(\mathcal{I})). \tag{6}$$

## 2.3 Data valuation vs Dataset valuation

A tentative, but naive, approach to solve the dataset valuation problem could be to run an auxiliary data-valuation algorithm on all the data and to assign to each dataset the sum of the values of its datapoints. We highlight the cons of this idea on a very simple, yet insightful example. Consider two datapoints $x_1$ and $x_2$, three datasets $\mathrm{D}_1 = \{x_1\}$, $\mathrm{D}_2 = \{x_2\}$, $\mathrm{D}_3 = \{x_2, x_2\}$, and the following toy utility function $u(\mathrm{D}) = \mathbb{1}\{x_1, x_2 \in \mathrm{D}\}$. In data valuation, any point $x_2$ shall have the same value, as they are identical. In particular, a naive summation would value $\mathrm{D}_3$ twice the value of $\mathrm{D}_2$. In dataset valuation, and for this toy problem at hand, it is quite clear that both datasets should have the same value. Moreover, the Shapley values are $1/6$ for $\mathrm{D}_2$ and $\mathrm{D}_3$ versus $2/3$ for $\mathrm{D}_1$.

The message here is twofold. Data valuation and dataset valuation are two fundamentally different concepts and one cannot directly reduce the latter to the former. This is actually true, and this is the second message, because the utility function $u$ is highly non-linear (even for the regression task).

# 3 Discrete Uniform Shapley Value

This section introduces and studies our approximation scheme for the Shapley value. Section 3.1 shows an asymptotic property that gives the general intuition behind our approximation. The result holds for the three use-cases of Sections 2.1.1 to 2.1.3. Section 3.2 presents a general approximation methodology for dataset valuation and shows its almost surely convergence as the number of players grows for our three uses-cases. Section 3.3 studies the rate of convergence, first for the homogeneous setting (Section 2.1.2), and then leverages this result to obtain a similar one for the non-parametric regression setting (Section 2.1.1). All proofs are postponed to the supplementary material.

## 3.1 Insights behind DU-Shapley

The Shapley value, by re-arranging the coalitions $\mathcal{S} \subseteq \mathcal{I} \setminus \{i\}$ by their cardinality in the sum in (5), can be equivalently expressed as

$$\varphi_i(u) = \mathbb{E}_{K \sim \mathrm{U}(\{0,\dots,I-1\})} \mathbb{E}_{\mathcal{S} \sim \mathrm{U}\left(2_K^{\mathcal{I} \setminus \{i\}}\right)} [u(\mathcal{S} \cup \{i\}) - u(\mathcal{S})], \tag{7}$$

where $2_K^{\mathcal{I} \setminus \{i\}}$ denotes the subsets of $\mathcal{I} \setminus \{i\}$ of cardinality $K$. In our three uses-cases, it follows that

$$\varphi_i(u) = \varphi_i(w) = \mathbb{E}_{K \sim \mathrm{U}(\{0,\dots,I-1\})} \mathbb{E}_{\mathcal{S} \sim \mathrm{U}\left(2_K^{\mathcal{I} \setminus \{i\}}\right)} [w(q(\mathcal{S} \cup \{i\})) - w(q(\mathcal{S}))], \tag{8}$$

where $w : \mathbb{R}_+ \to \mathbb{R}$ is such that $u(\mathcal{S}) = w(q(\mathcal{S}))$ for any $\mathcal{S} \subseteq \mathcal{I}$, and $q(\mathcal{S})$ is the scalar quantity of interest identified in Sections 2.1.1 to 2.1.3 for each use-case:

$$q(\mathcal{S}) := \left\lfloor \frac{\left(\sum_{i \in \mathcal{S}} \gamma_i n_i\right)^2}{\sum_{i \in \mathcal{S}} \gamma_i^2 n_i} \right\rfloor, \text{ where, for any } i \in \mathcal{I}, \gamma_i = \left\{ \begin{array}{ll} 1 & \text{for the second use-case,} \\ \sigma_i/\varepsilon_i & \text{for the third use-case,} \end{array} \right. \tag{9}$$

and for the first use-case, $q_b(\mathcal{S})$ is analogously defined at every bin, with $\gamma_i^b = 1$ for all players and all bins. We remark that the definition of $q(\mathcal{S})$ in the first and second use-cases is not restricted to linear regression. Equation (8) explicitly reveals a key random variable, namely $q(\mathcal{S})$. Interestingly, Figure 1 suggests that $q(\mathcal{S})$ converges in distribution to a uniform random variable as the number of players increases (with i.i.d. datasets sizes). Theorem 2 proves this result formally for any $(\gamma_i)_{i \in \mathcal{I}}$.

**Theorem 2.** *Let $\{n_i, \gamma_i\}_{i \in [I]}$ be two sequences of positive numbers such that the following limits*

$$\lim_{I \to \infty} \frac{1}{I} \sum_{i \in [I]} n_i \gamma_i = \mu_A, \quad \lim_{I \to \infty} \frac{1}{I} \sum_{i \in [I]} (n_i \gamma_i - \mu_A)^2 = \sigma_A^2,$$

$$\lim_{I \to \infty} \frac{1}{I} \sum_{i \in [I]} n_i \gamma_i^2 = \mu_B, \quad \lim_{I \to \infty} \frac{1}{I} \sum_{i \in [I]} (n_i \gamma_i^2 - \mu_B)^2 = \sigma_B^2,$$

*all exist, for some constants $\mu_A, \mu_B, \sigma_A, \sigma_B > 0$. Let $K \sim \mathrm{U}(\{0, \ldots, I\})$, $\mathcal{S}_K \sim \mathrm{U}([2_K^{\mathcal{I}}])$. Then, almost surely, $\frac{q(\mathcal{S}_K)}{q(\mathcal{I})} \xrightarrow{I \to \infty} \mathrm{U}([0,1])$.*

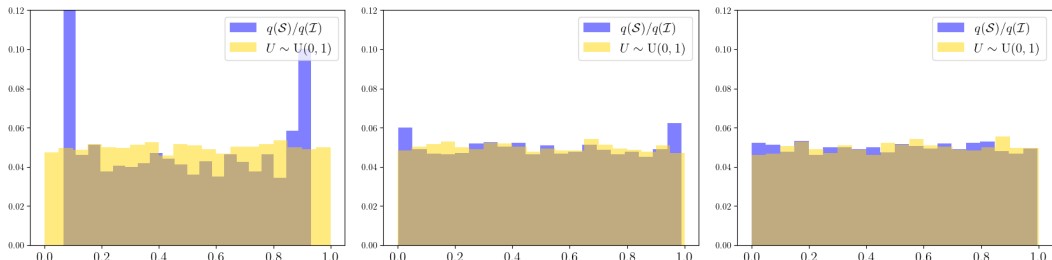

Figure 1: Distribution of $q(\mathcal{S})/q(\mathcal{I})$ when $\mathcal{S}$ is sampled as in (8) (i.e., first sample a size $K$ uniformly, then sample a coalition $\mathcal{S}$ of size $K$ uniformly). (left) $I = 10$, (middle) $I = 50$, (right) $I = 500$. We considered $10^4$ samples for each random variable, and the third use-case with $n_i \sim \mathrm{U}([100])$ and $\sigma_i/\varepsilon_i \sim \mathrm{U}([10])$ for each $i \in \mathcal{I}$.

## 3.2 Discrete Uniform Shapley value

The Shapley value re-arrangement in (7) exposes the main tool behind our approximation: it is enough to approximate the distribution of the random variable $\mathrm{D}_{\mathcal{S}}$ that takes values on the subsets of $\mathrm{D}_{-i} := \cup_{j \in \mathcal{I} \setminus \{i\}} \mathrm{D}_j$ (recall that $u(\mathcal{S}) = u(\mathrm{D}_{\mathcal{S}})$). Theorem 2, taking the example of the second use-case for intuition, indicates that these datasets have uniformly distributed numbers of points in the limit. Generalizing this intuition, we propose to approximate $\mathrm{D}_{\mathcal{S}}$ by taking $I$ samples of increasing size from the pool $\mathrm{D}_{-i}$ by sampling data points uniformly. This leads to the following definition of `DU-Shapley` for our generic model:

**Definition 3.** *[`DU-Shapley`] For any $i \in \mathcal{I}$, the discrete uniform Shapley value (`DU-Shapley`) of the $i$-th player, denoted by $\psi_i$, is given by*

$$\psi_i(u) := \frac{1}{I} \sum_{k=0}^{I-1} u(\mathrm{D}^{(k)} \cup \mathrm{D}_i) - u(\mathrm{D}^{(k)}),$$

*where $\mathrm{D}^{(k)}$ is a set of data points uniformly sampled without replacement from $\mathrm{D}_{-i}$ of size $k\mu_{-i}$, with $\mu_{-i} = \frac{1}{(I-1)}|\mathrm{D}_{-i}|$.*

Compared to the Shapley value defined in (5), which involves $2^I$ terms to compute, note that `DU-Shapley` only involves $I$ terms and hence it presents an exponential reduction of the number of utility function evaluations. Of course, these computational savings come at the cost of some bias. The latter is precisely quantified in Section 3.3 for our first two use-cases.

By definition, `DU-Shapley` is a random variable which depends on the sampled data points. However, whenever $u(\mathcal{S}) = w(q(\mathcal{S}))$, with $q(\mathcal{S})$ some scalar quantify of interest, as in our use-cases, we

can get rid of the stochastic nature of `DU-Shapley` by considering $I$ real values from well chosen intervals. In particular, in our uses-cases, `DU-Shapley` boils down to:

$$\psi_i(w) = \frac{1}{I} \sum_{k=0}^{I-1} w(\bar{q}_i^k) - w(\bar{q}_{-i}^k), \tag{10}$$

where

$$\bar{q}_i^k := \left\lfloor \frac{(\gamma_i n_i + \frac{k}{I-1} \sum_{j \in \mathcal{I} \setminus \{i\}} \gamma_j n_j)^2}{\gamma_i^2 n_i + \frac{k}{I-1} \sum_{j \in \mathcal{I} \setminus \{i\}} \gamma_j^2 n_j} \right\rfloor \text{ and } \bar{q}_{-i}^k := \left\lfloor \frac{k}{I-1} \cdot \frac{(\sum_{j \in \mathcal{I} \setminus \{i\}} \gamma_j n_j)^2}{\sum_{j \in \mathcal{I} \setminus \{i\}} \gamma_j^2 n_j} \right\rfloor.$$

We remark the notation abuse as we should write $\psi(w \circ q)$. For simplicity, we omit the composition and only write $\psi(w)$. Equation (10) coincides exactly with Definition 3 in the first two use-cases, i.e., when $\gamma_j = \gamma$ for all $j \in \mathcal{I}$. Indeed, as the random datasets $\mathrm{D}^{(k)}$ have a fixed size and the value function only looks at the number of data points within the coalition, we obtain,

$$\psi_i(u) = \frac{1}{I} \sum_{k=0}^{I-1} u(\mathrm{D}^{(k)} \cup \mathrm{D}_i) - u(\mathrm{D}^{(k)}) = \frac{1}{I} \sum_{k=0}^{I-1} w(|\mathrm{D}^{(k)} \cup \mathrm{D}_i|) - w(|\mathrm{D}^{(k)}|)$$

$$= \frac{1}{I} \sum_{k=0}^{I-1} w(k\mu_{-i} + n_i) - w(k\mu_{-i}) = \psi_i(w).$$

For the third use-case, Equation (10) is an approximation that comes from assuming that, for any $j \in \mathcal{I} \setminus \{i\}$, $|\mathrm{D}_j \cap \mathrm{D}^{(k)}| = k \cdot \frac{n_j}{I-1}$, which holds with high probability for large values of $I$, since

$$q(\mathrm{D}^{(k)} \cup \mathrm{D}_i) = \left\lfloor \frac{(\gamma_i n_i + \sum_{j \in \mathcal{I} \setminus \{i\}} \gamma_j \cdot |\mathrm{D}_j \cap \mathrm{D}^{(k)}|)^2}{\gamma_i^2 n_i + \sum_{j \in \mathcal{I} \setminus \{i\}} \gamma_j^2 \cdot |\mathrm{D}_j \cap \mathrm{D}^{(k)}|} \right\rfloor.$$

Theorem 2 implies the following result.

**Corollary 4.** *Let $\varphi_i$ and $\psi_i$ be, respectively, the Shapley value (5) and the `DU-Shapley` (10) of player $i$. Then, in our three uses-cases, it holds, $\lim_{I \to \infty} |\varphi_i - \psi_i| = 0$ almost surely.*

While our theoretical results are based on Equation (10) for the cases where $u(\mathcal{S}) = w(q(\mathcal{S}))$, we will see through numerical experiments that Definition 3 gives good results in more general cases.

### 3.3 Non-Asymptotic Theoretical Guarantees

Corollary 4 states asymptotic guarantees for `DU-Shapley`. In this section, we show non-asymptotic results that give the convergence rate for the first two uses-cases.[1] Recall that in non-parametric estimation, the utility writes as $u(\mathcal{S}) = \sum_{b \in [B]} u_b(\mathcal{S}) \mathbb{P}(x \in b)$, and therefore, by the linearity axiom of the Shapley value, for any $i \in \mathcal{I}$, $\varphi_i(u) = \sum_{b \in [B]} \varphi_i(u_b) \mathbb{P}(x \in b)$. As a consequence, in order to estimate $\varphi_i(u)$, it is enough to compute each $\varphi_i(u_b)$. In particular, the Shapley value approximation error over the whole feature space becomes a simple aggregation of the Shapley value approximation errors over the bins. We focus firstly on bounding the bias of our method in the homogeneous use-case to then extend it to the non-parametric regression case.

As in the homogeneous use-case the utility function writes as $u(\mathcal{S}) = w(\sum_{i \in \mathcal{I}} n_i)$, we consider the following regularity assumptions on $w$.

**H1.** *The function $w : \mathbb{R}_+ \to \mathbb{R}$ is increasing, twice continuously differentiable, and such that $\lim_{n \to \infty} n^2 |w^{(2)}(n)| < \infty$ (where $w^{(2)}$ represents the second derivative).*

Monotonicity is a natural assumption in our framework as, the more data, the more precise the ML prediction is expected to be. The condition over the limit aims at controlling the growth behavior of the utility function and it is automatically satisfied whenever $w$ is bounded and $w^{(2)}$ is monotone, by the mean value theorem. Theorem 5 bounds the bias of `DU-Shapley` for the homogeneous use-case.

**Theorem 5.** *Under Assumption H1, there exists a constant $\kappa > 0$, such that, for any $i \in \mathcal{I}$, it holds,*

$$\left| \varphi_i - \psi_i \right| \leq \frac{\kappa}{(I-1)\mu_{-i}^2} \left( \sigma_{-i}^2 (1 + \ln(I-1)) + \zeta_{-i} \right),$$

---

[1]A similar result, albeit more technical, can be shown with the same arguments for the third use-case.

where $\varphi_i$ and $\psi_i$ are respectively the Shapley value and the `DU-Shapley` of player $i$, $\mu_{-i} = \frac{1}{(I-1)}|\mathrm{D}_{-i}|$ is the average dataset size of all players but $i$, $\sigma^2_{-i} = \frac{1}{I-1}\sum_{j\in\mathcal{I}\setminus\{i\}}(n_j - \mu_{-i})^2$ their empirical variance, and $\zeta_{-i}$ measures the variability of the dataset sizes across players. Formally, it is defined as $\zeta_{-i} := R^2_{-i}\tau^2_{-i}/4n^{\max}_{-i}$ where $R_{-i} := \max_{j\in\mathcal{I}\setminus\{i\}}|n_j - \mu_{-i}|$, $n^{\max}_{-i} := \max_{j\in\mathcal{I}\setminus\{i\}} n_j$, and $\tau_{-i} := n^{\max}_{-i}/\min_{j\in\mathcal{I}\setminus\{i\}} n_j$.

The full proof of Theorem 5 is included in Appendix C.3 and it relies on controlling the absolute value of $\mathbb{E}[w(\mu_{-i}K) - w(\sum_{j\in\mathcal{S}} n_j)]$, where $K \sim \mathrm{U}(\{0, ..., I-1\})$ and $\mathcal{S}$ is the random variable in (7). Using a second order Taylor expansion, the problem is reduced to controlling the term related to the second derivative of $w$ by using the regularity assumptions in **H**1.

As advertised before, Theorem 5 can be directly generalized to the non-parametric use-case, since,

$$u(\mathcal{S}) = \sum_{b\in[B]} w_b\left(\sum_{i\in\mathcal{S}} n_{i,b}\right)\mathbb{P}(x \in b), \text{ for } n_{i,b} = |\{(x,y) \in \mathrm{D}_j, x \in b\}|.$$

**Corollary 6.** *Under Assumption **H**1 for all functions $w_b$, there exist constants $\kappa_b > 0$, such that, for any $i \in \mathcal{I}$, it holds that*

$$|\varphi_i - \psi_i| \leq \sum_{b\in[B]} \frac{\kappa_b \mathbb{P}(x \in b)}{(I-1)\mu^2_{-i,b}}\left(\sigma^2_{-i,b}(1 + \ln(I-1)) + 2\zeta_{-i,b}\right), \tag{11}$$

*where $\varphi_i$ and $\psi_i$ are respectively the Shapley value and the `DU-Shapley` of player $i$, and all terms are equivalently defined to Theorem 5 at each bin $b \in [B]$.*

The upper bound in (11) depends on natural quantities related to the dataset valuation problem described in Section 2.1 at each bin, such as the first two moments $\mu_{-i,b}$ and $\sigma_{-i,b}$ of the datasets' size distribution. More precisely, the error increases when there are some outlier players with a very small or large dataset size. This behavior is expected since, in this particular setting, the random variable inside of the Shapley value differs from a uniform random variable. As showcased in Theorem 2, the error vanishes when the number of players $I$ tends towards infinity.

## 4 Numerical Experiments

We illustrate the benefits of `DU-Shapley` by measuring numerically three properties: (1) how well `DU-Shapley` approximates the Shapley value in real data, (2) how many (theoretical) iterations need other methods to achieve the same accuracy level than `DU-Shapley`, and (3) how well `DU-Shapley` performs in classical dataset valuation tasks with real data. Appendix A.1 complements the results by a complexity comparison between our method and `SVARM` [22] and Appendix A.2 by experiments on synthetic data. The experiments strongly suggest that `DU-Shapley` performs well in all tasks.

### 4.1 Approximating the Shapley Value in Real-World Data

We consider the real-world datasets in Mitchell et al. [32], whose details are provided in Table 3 in the appendix. To tackle these problems we consider logistic regression models and gradient-boosted decision trees (GBDT). For classification tasks, the utility function has been taken as the expected accuracy of the trained logistic regression model over a hold-out testing set while for regression tasks, the utility function corresponds to the averaged MSE over a hold-out testing set. In both cases we took a hold-out testing set with 10% of the size of the training dataset. For each dataset, we considered two worst-case scenarios for our method, namely $I = 10$ players and $I = 20$ players.

Starting from the datasets in Table 3, we heterogeneously allocate datasets to the players. We compare ourselves with two approaches, referred to as `MC-Shapley`, for the standard MC approximation defined in (6), and `MC-anti-Shapley` that considers, in addition, antithetic sampling [32]. We compute the averaged MSE across all players between the true Shapley value and each estimator.

Since computing the marginal contributions in this experiment requires re-training, which is clearly not feasible for a large number of epochs, we chose to restrict ourselves to 20 steps of stochastic gradient descent for logistic regression and 20 boosting iterations for GBDTs. For MC-based approaches, we considered $I$ samples to compare those approximations with the proposed methodology on a fair basis, i.e., associated to the same computational budget.

Table 1 depicts the results. We clearly see that, even in the worst-case scenario where the number of players is small and far from the theoretical assumptions from Section 3.3, `DU-Shapley` competes favorably with the MC-based methods.

Table 1: Worst-case comparison between `DU-Shapley` and competitors, for real-world datasets considered in Table 3. We report the averaged MSE across all players w.r.t. the exact Shapley value.

| Dataset | adult | | breast-cancer | | bank | | cal-housing | |
|---|---|---|---|---|---|---|---|---|
| Players | 10 | 20 | 10 | 20 | 10 | 20 | 10 | 20 |
| DU-Shapley | $\mathbf{2.10^{-3}}$ | $\mathbf{6.10^{-4}}$ | $\mathbf{3.10^{-3}}$ | $\mathbf{1.10^{-4}}$ | $\mathbf{5.10^{-2}}$ | $\mathbf{4.10^{-3}}$ | $\mathbf{1.10^{-2}}$ | $\mathbf{3.10^{-3}}$ |
| MC-Shapley | $1.10^{-2}$ | $4.10^{-3}$ | $3.10^{-2}$ | $1.10^{-3}$ | $9.10^{-2}$ | $6.10^{-2}$ | $5.10^{-2}$ | $2.10^{-2}$ |
| MC-anti-Shapley | $8.10^{-3}$ | $2.10^{-3}$ | $1.10^{-2}$ | $8.10^{-4}$ | $8.10^{-2}$ | $4.10^{-2}$ | $3.10^{-2}$ | $1.10^{-2}$ |

| Dataset | make-regression | | year | |
|---|---|---|---|---|
| Players | 10 | 20 | 10 | 20 |
| DU-Shapley | $\mathbf{9.10^{-2}}$ | $\mathbf{2.10^{-2}}$ | $\mathbf{1.10^{-3}}$ | $\mathbf{7.10^{-4}}$ |
| MC-Shapley | $4.10^{-1}$ | $3.10^{-1}$ | $5.10^{-3}$ | $1.10^{-3}$ |
| MC-anti-Shapley | $4.10^{-1}$ | $2.10^{-1}$ | $5.10^{-3}$ | $1.10^{-3}$ |

## 4.2 Complexity of Computing the Shapley Values of all Players

We have looked at the number of iterations that `DataShapley` and the *Improved Group Testing-Based* method [46] (`IGTB`) require to achieve `DU-Shapley`'s accumulated bias, formally given by

$$\mathrm{DUbias}(I) := \frac{\kappa}{I-1} \left( \sum_{i \in \mathcal{I}} \frac{(9\sigma^2_{-i}(1 + \log(I-1)) + \zeta_{-i})^2}{(\mu_{-i})^4} \right)^{1/2}.$$

To do so, we have replaced $\varepsilon = \mathrm{DUbias}(I)$, respectively, in the formula in Section 4.1 in [16] and Equation 5 in [46], with a value function motivated from our third use-case under the homogeneity assumption $\sigma_i/\varepsilon_i = \sigma/\varepsilon$ for all $i \in \mathcal{I}$. The results are illustrated in Figure 2. Remark `DU-Shapley` requires $I^2$ iterations to compute all Shapley values. We observe that in all tested instances, both methods require a higher number of iterations to achieve the same error than DU-Shapley.

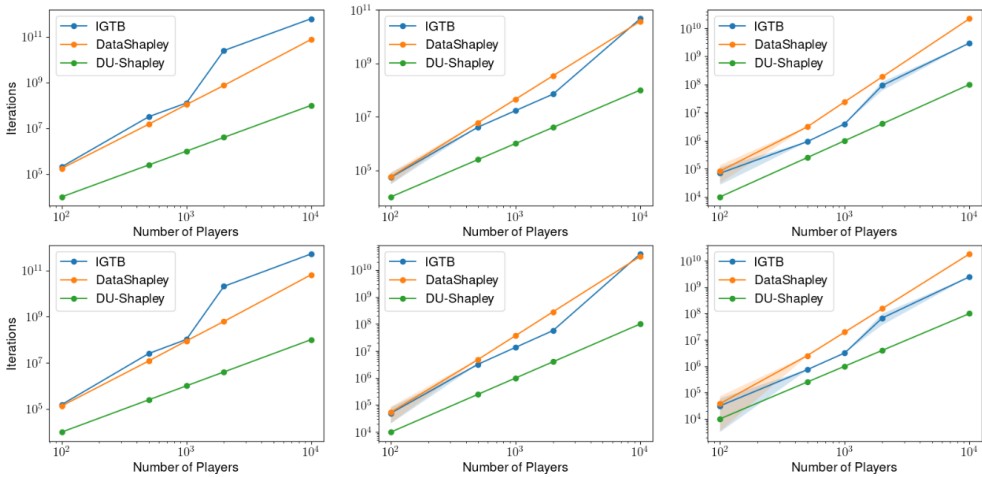

Figure 2: Iterations required by DataShapley and the Improved Group Testing-Based method to achieve DU-Shapley's accumulated bias with function $w(n_S) = 1 - \frac{10^{k(\mathcal{I})}}{10^{k(\mathcal{I})}+n_S}$, where $n_S$ is the number of data points of the coalition $S \subseteq \mathcal{I}$, and $k(\mathcal{I}) := \lfloor \log(\sum_{i \in \mathcal{I}} n_i) \rfloor - 1$ is a normalization factor. (top) $\delta = 0.01$, (bottom) $\delta = 0.1$, (left) $n_{\max} = 10$, (middle) $n_{\max} = 50$, (right) $n_{\max} = 100$.

## 4.3 Applying `DU-Shapley` to dataset valuation problems

We considered non-tabular datasets used in [17], namely bbc-embedding, IMDB-embedding, both text datasets, and CIFAR10-embedding, an image dataset. Feature embedding have been generated

using pretrained DistilBERT and ResNet50 models, respectively. In addition we have adapted three baselines from data valuation to our setting: Leave-One-Out (LOO), DataShapley, and KNN-Shapley. Appendix B.2 gives the implementations details. For these datasets associated to classification problems, we used a multi-layer perceptron classifier as prediction model.

We have considered three dataset valuation problems, none of them needing the real Shapley values, which allows us to increase the number of players w.r.t. the experiments in Section 4.1. We investigated noisy label detection (NLD), dataset removal (DR), and dataset addition (DA) [17]. For NLD, we used as a metric the F1-score (the larger the better). For DR, we used the testing accuracy (the lesser the better). For DA, we used the testing accuracy (the lesser the better).

We considered splitting the dataset across $I = 100$ players. The results are summarized in Table 2. We observe that `DU-Shapley` has competitive results compared to classical baselines despite of the fact that none of the considered cases verifies the structural assumptions from Section 3.3. In addition, we can see that `DU-Shapley` tends to have similar and even better results as `Data Shapley` (which is a MC based method). This is in line with our theory as, for larger number of players, `DU-Shapley` tends to better estimate the true Shapley value.

Table 2: Comparison between `DU-Shapley` and competitors for real-world datasets considered in [17] in Noisy label detection, Dataset Removal and Dataset Addition.

| Dataset | CIFAR 10 | | | | | | BBC | | | | | |
|---|---|---|---|---|---|---|---|---|---|---|---|---|
| Problem | NLD | | DR | | DA | | NLD | | DR | | DA | |
| | 5% | 15% | 5% | 15% | 5% | 15% | 5% | 15% | 5% | 15% | 5% | 15% |
| Random | 0.11 | 0.19 | 0.61 | 0.60 | 0.25 | 0.41 | 0.11 | 0.19 | 0.90 | 0.88 | 0.68 | 0.81 |
| LOO | 0.13 | 0.18 | 0.62 | 0.60 | 0.15 | 0.32 | 0.11 | 0.17 | 0.90 | 0.88 | 0.61 | 0.77 |
| DataShapley | 0.13 | 0.25 | 0.61 | 0.59 | 0.12 | 0.18 | 0.12 | 0.20 | 0.89 | 0.87 | 0.08 | 0.12 |
| KNN-Shapley | **0.14** | 0.28 | **0.60** | 0.57 | 0.12 | 0.15 | **0.19** | 0.29 | **0.88** | 0.86 | 0.13 | 0.12 |
| DU-Shapley | **0.14** | **0.30** | 0.61 | **0.55** | **0.11** | **0.14** | 0.18 | **0.34** | 0.89 | **0.85** | **0.07** | **0.11** |

| Dataset | IMBD | | | | | |
|---|---|---|---|---|---|---|
| Problem | NLD | | DR | | DA | |
| | 5% | 15% | 5% | 15% | 5% | 15% |
| Random | 0.10 | 0.16 | 0.77 | 0.75 | 0.62 | 0.68 |
| LOO | 0.11 | 0.18 | 0.77 | 0.74 | 0.53 | 0.59 |
| DataShapley | 0.17 | 0.28 | **0.75** | 0.69 | 0.36 | **0.33** |
| KNN-Shapley | **0.18** | 0.29 | 0.76 | 0.68 | 0.41 | 0.37 |
| DU-Shapley | **0.18** | **0.32** | 0.76 | **0.66** | **0.33** | 0.34 |

## 5 Conclusion

We model the dataset valuation problem as a cooperative game and design a Shapley value approximation, named `DU-Shapley`, that exploits the underlying structure of the utility function and exponentially reduces the number of functions valuations required for the computation. In three different uses-cases, `DU-Shapley` is proved to almost surely converge to the real Shapley value as the number of players grows. Moreover, we find the rate of convergence, which depends only on natural parameters of dataset valuation. Numerical experiments showcase that `DU-Shapley` performs well in approximating the Shapley value and performing dataset valuation tasks, even when the assumptions needed for the theoretical guarantees do not hold, and it has a good complexity when computing the Shapley values of all players.

**Limitations of our method**. Our non-asymptotic bound for the non-parametric regression setting in Corollary 6 indicates that `DU-Shapley` works better when agents' datasets are *regular* in the sense that they have similar sizes. Hence, a limitation of our approximation is that it may work poorly in settings where some players have large datasets compared to others, as the distribution of the random variable within the Shapley value drives apart from being uniform. Moreover, our convergence result in Theorem 2 (for all use-cases) assume the existence of limits, which roughly requires that heterogeneity between players—in terms of both dataset size and variance—can be bounded. This also indicates that convergence may be not be guaranteed if the heterogeneity is arbitrarily high.

## Acknowledgments

This research was supported in part by the French National Research Agency (ANR) in the framework of the PEPR IA FOUNDRY project (ANR-23-PEIA-0003) and through the grant DOOM ANR-23-CE23-0002. It was also funded by the European Union (ERC, Ocean, 101071601). Views and opinions expressed are however those of the author(s) only and do not necessarily reflect those of the European Union or the European Research Council Executive Agency. Neither the European Union nor the granting authority can be held responsible for them.

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

# A Complementary Numerical results

All experiments were executed on a laptop running macOS 13.3.1 and equipped with Apple M1 chip with 16GB of RAM. The minimum amount of compute was roughly 5 minutes while the maximum one roughly 10 hours.

## A.1 DU-Shapley vs SVARM

We have looked at the probability at which SVARM (Theorem 4 [22]) can ensure, after $I^2$ iterations (without considering the warm up as part of the budget), an error equal to DU-Shapley's accumulated bias. We have considered the same value function than in Section 4.2 with $n_{max} \in \{2 \cdot 10^3, 3 \cdot 10^3, 5 \cdot 10^3, 10^4\}$ and 100 simulations of sets of players at each time. Figure 3 shows the results. We observe how SVARM cannot ensure, with high enough probability, an approximation error equal to the one of DU-Shapley.

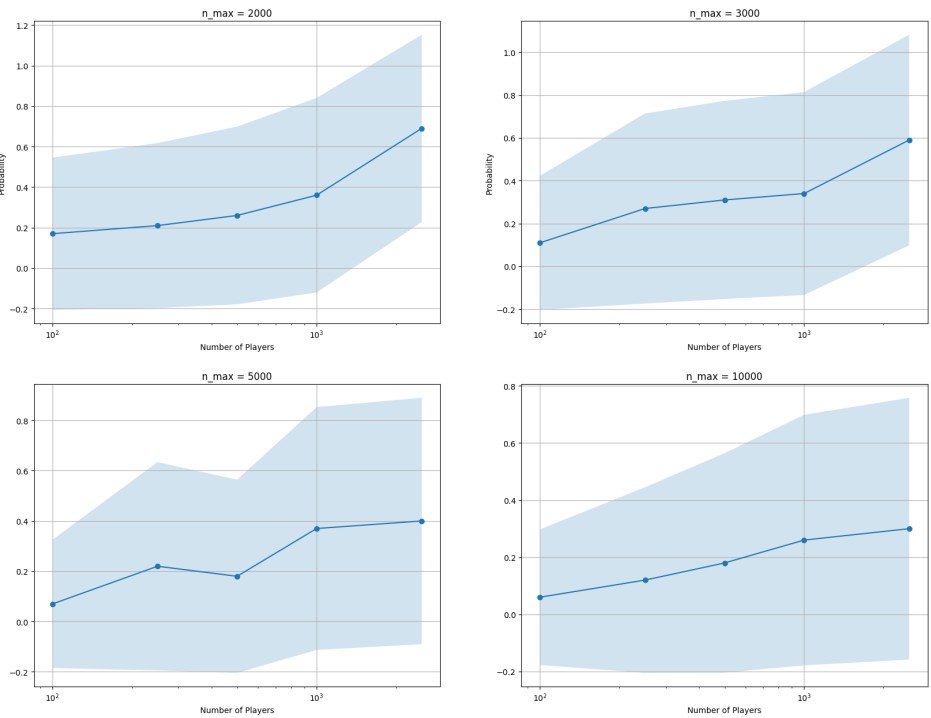

Figure 3: Probability that SVARM guarantees an error equal to DU-Shapley's bias

## A.2 Approximating the Shapley value in Synthetic Data

We consider a toy dataset valuation problem associated to our heterogeneous linear regression with local differential privacy use-case (Section 2.1.3) and we measure the value of a coalition $\mathcal{S}$ with the utility function in close-form from Proposition 1. We consider $d = 10$.

In order to benchmark the performances of `DU-Shapley`, we consider four competitive approaches, relying on Monte Carlo (MC) approximation strategies [32]. The first one, referred to as `MC-Shapley` is the standard MC approximation defined in (6). The second one, coined `MC-anti-Shapley` is a variance-reduced version of `MC-Shapley` that considers antithetic sampling. The third one coined `Owen-Shapley` stands for the multilinear extension of Owen [34] which represents the Shapley value as two nested expectations (further explained in Appendix B.3). Finally, the fourth approach, coined `Orthogonal-Shapley`, relies on efficient permutation sampling techniques on the hypersphere to draw permutations in (4) in a dependent way. To assess the performance of the aforementioned Shapley value estimators, we used the mean square error (MSE) averaged over all players. `DU-Shapley` is computed exactly by using (10) while, for each MC-based estimator, we performed 25 Shapley value estimations to compute the MSE, and did it 10 times to obtain confidence intervals for the MSE.

Figure 4 compares `DU-Shapley` (the horizontal line which does not depend on the sampling budget as we compute it exactly) and the MC-based methods, which are computed at several different budgets. The x-axis corresponds to the sampling budget allowed to the MC-bases methods w.r.t. `DU-Shapley`, i.e., $10^{-1}$ means a budget equal to 10% the one of `DU-Shapley`, $10^0$ means same budget (indicated by the black vertical line), and $10^1$ means 10 times the `DU-Shapley` budget. Remark that, even when the MC-methods use 10 times the budget of `DU-Shapley`, our method keeps approximating better the Shapley value.

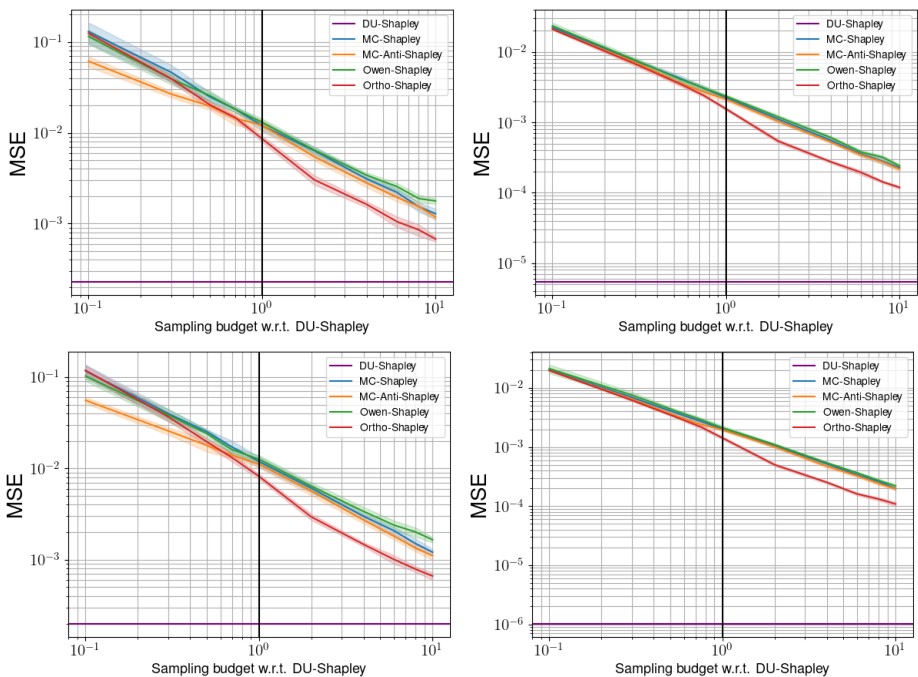

Figure 4: Worst-case comparison between $\texttt{DU-Shapley}$ and MC-based approximations with different budgets on synthetic datasets. From left to right, $I = 10$ and $I = 20$, $n_i \sim \mathrm{U}([10^3]), \forall i \in \mathcal{I}$. (top) Scenario with small heterogeneity, $\sigma_i/\varepsilon_i \sim \mathrm{U}([0, 10]), \forall i \in \mathcal{I}$, (bottom) scenario with high heterogeneity, $\sigma_i/\varepsilon_i \sim \mathrm{U}([0, 100]), \forall i \in \mathcal{I}$.

## B   Further details about numerical implementations

### B.1   Datasets considered in Section 4.1.

Table 3 summarizes the real-world datasets considered in Section 4.1.

Table 3: Datasets considered in Section 4.1.

| Dataset | Size | $d$ | Task |
|---|---|---|---|
| adult [21] | 48,842 | 107 | classification |
| breast-cancer [30] | 699 | 30 | classification |
| bank [33] | 45,211 | 16 | classification |
| cal-housing [19] | 20,640 | 8 | regression |
| make-regression [36] | 1,000 | 10 | regression |
| year [36] | 515,345 | 90 | regression |

### B.2   OpenDataVal implementations

In this section we describe more in detail the implementations of DataShapley, Leave-One-Out (LOO), and KNN-Shapley for our numerical results in Section 4.3.

DataShapley, applied to the dataset valuation problem, simply corresponds to the method coined MC in Section 4.1. Therefore, we sample datasets and output the averaged marginal contribution.

Regarding LOO, notice that it corresponds to compute just one marginal contribution, usually computed on the big coalition, i.e.,

$$\mathrm{LOO}_i := u(\mathcal{I}) - u(\mathcal{I} \setminus \{i\}).$$

As players' marginal contributions to large datasets tend to be small, we have preferred to sample one dataset D from $D_{-i}$ and to output

$$\text{LOO}_i := u(D \cup D_i) - u(D).$$

Finally, regarding KNN-Shapley, we refer the reader to [15], Section E.3 of the appendix who explain how to adapt the method to dataset valuation.

### B.3 Owen's Shapley value approximation

In Section A.2, we considered the Shapley value approximation referred to as `Owen-Shapley` as a state-of-the-art competitor to `DU-Shapley`. We provide in the following additional details regarding `Owen-Shapley`. For the other competitors, we directly refer the interested reader to Mitchell et al. [32]. Owen [34] studied the multilinear extension of a cooperative game and an alternative way to express the Shapley value. Formally, a cooperative game $G = (\mathcal{I}, u)$ consists on a set of $I$ players $\mathcal{I} = \{1, 2, ..., I\}$ and a value function $u : 2^{\mathcal{I}} \to \mathbb{R}$ such that, for any $S \subseteq \mathcal{I}$, $u(S)$ corresponds to the value generated by the coalition $S$. The multilinear extension of $G$, denoted $\bar{G} = (\mathcal{I}, \bar{u})$, is obtained when considering the value function $\bar{u} : [0, 1]^{\mathcal{I}} \to \mathbb{R}$ given by,

$$\bar{u}(x_1, x_2, ..., x_I) = \sum_{S \subseteq \mathcal{I}} \prod_{i \in S} x_i \prod_{j \notin S} (1 - x_i) u(S).$$

Intuitively, $\bar{u}(x_1, x_2, ..., x_I)$ corresponds to the expected value of a coalition when each player $i \in \mathcal{I}$ joins the coalition with probability $x_i$. Theorem 5 in [34] gives an alternative way to compute the Shapley value $\varphi_i(u)$ of player $i$ in game $G$, namely,

$$\varphi_i(u) = \int_0^1 \frac{\partial \bar{u}}{\partial x_i}(\tau, ..., \tau) d\tau = \int_0^1 \sum_{S \subseteq \mathcal{I} \setminus \{i\}} \tau^{|S|} (1 - \tau)^{I - |S| - 1} [u(S \cup \{i\}) - u(S)] d\tau$$

$$= \int_0^1 \mathbb{E}[u(\mathcal{E}_i(\tau) \cup i) - u(\mathcal{E}_i(\tau))] d\tau = \mathbb{E}_{\tau \sim U([0,1])}\left[\mathbb{E}[u(\mathcal{E}_i(\tau) \cup i) - u(\mathcal{E}_i(\tau))]\right],$$

where $\mathcal{E}_i(\tau)$ is a random subset of $\mathcal{I} \setminus \{i\}$, such that, $\forall j \in \mathcal{I} \setminus \{i\}$, $j$ is included in $\mathcal{E}_i(\tau)$ with probability $\tau$. In words, the Shapley value of player $i$ corresponds to her expected marginal contribution to the random set $\mathcal{E}_i(\tau)$, when $\tau$ is uniformly distributed on $[0, 1]$. This brings an alternative way to use Monte Carlo to approximate the Shapley value $\varphi_i(u)$, coined Owen-Shapley, as,

$$\hat{\varphi}_i^{\text{Owen}}(u) = \frac{1}{T} \sum_{t=1}^{T} u(\mathcal{E}_i(\tau_t) \cup i) - u(\mathcal{E}_i^t(\tau_t)),$$

where for each $t \in \{1, ..., T\}$, we draw $\tau_t$ independently and uniformly in $[0, 1]$ and then, create a random set $\mathcal{E}_i(\tau_t)$ by adding each player $j \in \mathcal{I} \setminus \{i\}$ to it with probability $\tau_t$.

## C   Missing proofs

### C.1   Proof of Proposition 1

**Proposition 1.** *Let $\mathcal{S} \subseteq \mathcal{I}$ be a coalition of players and consider the value function $u$ as in (3). It follows,*

$$u(\mathcal{S}) = \frac{-\text{Tr}[\mathbb{E}[xx^\top]]}{q(\mathcal{S}) - d - 1}, \text{ where } q(\mathcal{S}) := \left| \frac{\left(\sum_{i \in \mathcal{S}} \frac{\sigma_i}{\varepsilon_i} n_i\right)^2}{\sum_{i \in \mathcal{S}} \left(\frac{\sigma_i}{\varepsilon_i}\right)^2 n_i} \right|, \text{ with the convention } q(\varnothing) = 0.$$

*In particular, considering $p_{\text{test}} = N(0, I_d)$, we get,*

$$u(\mathcal{S}) = \frac{d}{d + 1 - q(\mathcal{S})}.$$

*Proof.* Let $\mathcal{S} \subseteq \mathcal{I}$ be a coalition of players and $X_\mathcal{S}, Y_\mathcal{S}$ be the concatenation of their datasets. The linear model can be rewritten in matrix form as

$$Y_\mathcal{S} = X_\mathcal{S}\theta + \eta_\mathcal{S},$$

where $\eta_\mathcal{S}$ is the concatenation of $\eta_i^{(j)}$ for all $i \in \mathcal{S}$ and $j \in [n_i]$. Take $\hat{\theta}_\mathcal{S} = (X_\mathcal{S}^\top \Sigma_\mathcal{S}^{-1} X_\mathcal{S})^{-1} X_\mathcal{S}^\top \Sigma_\mathcal{S}^{-1} Y_\mathcal{S}$ where $\Sigma_\mathcal{S} = \mathrm{diag}((\varepsilon_i^2)_{i\in\mathcal{S}})$, and let $x \sim p_{\text{test}}$ be a hold-out testing datum in $\mathbb{R}^d$. It follows,

$$
\begin{aligned}
\left(x^\top(\theta - \hat{\theta}_\mathcal{S})\right)^2 &= \left(\sum_{i\in\mathcal{S}} \eta_i \varepsilon_i^{-2} X_i\right)\left(\sum_{i\in\mathcal{S}} X_i^\top \varepsilon_i^{-2} X_i\right)^{-1} xx^\top \left(\sum_{i\in\mathcal{S}} X_i^\top \varepsilon_i^{-2} X_i\right)^{-1}\left(\sum_{i\in\mathcal{S}} \eta_i \varepsilon_i^{-2} X_i\right) \\
&= \mathrm{Tr}\left[\left(\sum_{i\in\mathcal{S}} \eta_i \varepsilon_i^{-2} X_i\right)\left(\sum_{i\in\mathcal{S}} X_i^\top \varepsilon_i^{-2} X_i\right)^{-1} xx^\top \left(\sum_{i\in\mathcal{S}} X_i^\top \varepsilon_i^{-2} X_i\right)^{-1}\left(\sum_{i\in\mathcal{S}} \eta_i \varepsilon_i^{-2} X_i\right)\right] \\
&= \mathrm{Tr}\left[xx^\top\left(\sum_{i\in\mathcal{S}} X_i^\top \varepsilon_i^{-2} X_i\right)^{-1}\left(\sum_{i\in\mathcal{S}} \eta_i \varepsilon_i^{-2} X_i\right)\left(\sum_{i\in\mathcal{S}} \eta_i \varepsilon_i^{-2} X_i\right)\left(\sum_{i\in\mathcal{S}} X_i^\top \varepsilon_i^{-2} X_i\right)^{-1}\right] \\
&= \mathrm{Tr}\left[xx^\top\left(\sum_{i\in\mathcal{S}} X_i^\top \varepsilon_i^{-2} X_i\right)^{-1}\left(\sum_{i\in\mathcal{S}}\sum_{j\in\mathcal{S}} X_i^\top \varepsilon_i^{-2} \eta_i \eta_j^\top \varepsilon_j^{-2} X_j\right)\left(\sum_{i\in\mathcal{S}} X_i^\top \varepsilon_i^{-2} X_i\right)^{-1}\right]
\end{aligned}
$$

We take expectation with respect to the different stochastic terms. Since $\eta_i^{(k)} \sim \mathrm{N}(0, \varepsilon_i^2)$ for any $i \in \mathcal{S}, k \in [n_i]$, it holds,

$$
\begin{aligned}
\mathbb{E}_{(\eta_i^{(k)} \sim \mathrm{N}(0,\varepsilon_i^2))_{i\in\mathcal{S}}^{k\in[n_i]}}&\left[\left(x^\top(\theta - \hat{\theta}_\mathcal{S})\right)^2\right] \\
&= \mathrm{Tr}\left[xx^\top\left(\sum_{i\in\mathcal{S}} X_i^\top \varepsilon_i^{-2} X_i\right)^{-1}\left(\sum_{i\in\mathcal{S}} X_i^\top \varepsilon_i^{-2} X_i\right)\left(\sum_{i\in\mathcal{S}} X_i^\top \varepsilon_i^{-2} X_i\right)^{-1}\right] \\
&= \mathrm{Tr}\left[xx^\top\left(\sum_{i\in\mathcal{S}} X_i^\top \varepsilon_i^{-2} X_i\right)^{-1}\right]
\end{aligned}
$$

Since players distributions differ on their variances, $\sum_{i\in\mathcal{S}} X_i^\top \varepsilon_i^{-2} X_i$ corresponds to a semi-correlated Wishart random variable where each $\frac{1}{\varepsilon_i} X_i \sim \mathrm{N}(0, (\frac{\sigma_i}{\varepsilon_i})^2 \mathrm{I}_d)$. In particular, the semi-correlated Wishart random variable can be approximated by a central Wishart distribution [37, 43], whose precision depends on the homogeneity of the coefficients $\sigma_i/\varepsilon_i$ over all $i \in \mathcal{I}$, as showed in [20]. It follows,

$$\mathbb{E}_{(X_i \sim \mathrm{N}(0,\sigma_i^2 \mathrm{I}_d))_{i\in\mathcal{S}}}\left[\left(\sum_{i\in\mathcal{S}} X_i^\top \varepsilon_i^{-2} X_i\right)^{-1}\right] \approx \frac{\mathrm{I}_d}{(q(\mathcal{S}) - d - 1)},$$

where

$$q(\mathcal{S}) := \left\lfloor \frac{\left(\sum\limits_{i\in\mathcal{S}} \frac{\sigma_i}{\varepsilon_i} n_i\right)^2}{\sum\limits_{i\in\mathcal{S}} \left(\frac{\sigma_i}{\varepsilon_i}\right)^2 n_i} \right\rfloor.$$

With all this in mind, it follows,

$$\mathbb{E}_{\substack{(\eta_i^{(j)} \sim \mathrm{N}(0,\varepsilon_i^2))_{i\in\mathcal{S}}^{j\in[n_i]} \\ (X_i \sim \mathrm{N}(0,\sigma_i^2 \mathrm{I}_d))_{i\in\mathcal{S}}}}\left[\left(x^\top(\theta - \hat{\theta}_\mathcal{S})\right)^2\right] = \mathrm{Tr}\left[xx^\top \frac{\mathrm{I}_d}{(q(\mathcal{S}) - d - 1)}\right] = \frac{1}{q(\mathcal{S}) - d - 1}\mathrm{Tr}\left[xx^\top\right].$$

In particular, considering $p_{\text{test}} = \mathrm{N}(0, \mathrm{I}_d)$, we get,

$$u(\mathcal{S}) = \frac{d}{d + 1 - q(\mathcal{S})}.$$

$\square$

## C.2 Proof of Theorem 2

**Theorem 2.** *Let $\{n_i, \gamma_i\}_{i \in [I]}$ be two sequences of positive numbers such that the following limits*

$$\lim_{I \to \infty} \frac{1}{I} \sum_{i \in [I]} n_i \gamma_i = \mu_A, \quad \lim_{I \to \infty} \frac{1}{I} \sum_{i \in [I]} (n_i \gamma_i - \mu_A)^2 = \sigma_A^2,$$

$$\lim_{I \to \infty} \frac{1}{I} \sum_{i \in [I]} n_i \gamma_i^2 = \mu_B, \quad \lim_{I \to \infty} \frac{1}{I} \sum_{i \in [I]} (n_i \gamma_i^2 - \mu_B)^2 = \sigma_B^2,$$

*all exist, for some constants $\mu_A, \mu_B, \sigma_A, \sigma_B > 0$. Let $\mathbf{K} \sim \mathrm{U}(\{0, \dots, I\})$, $\mathcal{S}_\mathbf{K} \sim \mathrm{U}([2^\mathcal{I}_\mathbf{K}])$, and define $q(\mathcal{S}_\mathbf{K})$ as in (9) for the third use-case. Then, almost surely, $q(\mathcal{S}_\mathbf{K})/q(\mathcal{I}) \xrightarrow{I \to \infty} \mathrm{U}([0, 1])$.*

*Proof.* Introduce, for any $t, t_0 \in (0, 1)$ and any $s > 0$,

$$\mu_A(I) = \frac{1}{I} \sum_{i \in [I]} n_i \gamma_i, \quad \mu_B(I) = \frac{1}{I} \sum_{i \in [I]} n_i \gamma_i^2,$$

$$Y_A(t, I) = \sum_{i \in \mathcal{S}_{\lfloor It \rfloor}} n_i \gamma_i, \quad Y_B(t, I) = \sum_{i \in \mathcal{S}_{\lfloor It \rfloor}} n_i \gamma_i^2.$$

$$R_A(I, t_0, s) = \mathbb{P} \left( \sup_{t > t_0} \left| \frac{Y_A(t, I)}{\lfloor It \rfloor} - \mu_A(I) \right| > s \right),$$

$$R_B(I, t_0, s) = \mathbb{P} \left( \sup_{t > t_0} \left| \frac{Y_B(t, I)}{\lfloor It \rfloor} - \mu_B(I) \right| > s \right).$$

By construction, $Y_A(t, I)$ and $Y_B(t, I)$ are sums of sampling without replacement of $\lfloor It \rfloor$ elements. Therefore, by Corollary 1.3 in [39], for $s$ fixed, there exists $I_0^A, I_0^B \in \mathbb{N}$ such that,

$$R_A(I, t_0, s) \le \frac{(1 - t_0)\sigma_A^2}{\lfloor It_0 \rfloor s^2}, \forall I \ge I_0^A \text{ and } R_B(I, t_0, s) \le \frac{(1 - t_0)\sigma_B^2}{\lfloor It_0 \rfloor s^2}, \forall I \ge I_0^B.$$

In other words, for any $s > 0$ and $I$ large enough, almost surely, it holds,

$$\left| \frac{Y_A(t, I)}{\lfloor It \rfloor} - \mu_A(I) \right| \le s \text{ and } \left| \frac{Y_B(t, I)}{\lfloor It \rfloor} - \mu_B(I) \right| \le s.$$

It follows,

$$\left| \frac{q(\mathcal{S}_{\lfloor It \rfloor})}{\lfloor It \rfloor} - \frac{\mu_A(I)^2}{\mu_B(I)} \right| = \left| \frac{1}{\lfloor It \rfloor} \cdot \frac{Y_A(t, I)^2}{Y_B(t, I)} - \frac{\mu_A(I)^2}{\mu_B(I)} \right|$$

$$= \left| \left( \frac{Y_A(t, I)^2}{\lfloor It \rfloor^2} - \mu_A(I)^2 + \mu_A(I)^2 \right) \left( \frac{\lfloor It \rfloor}{Y_B(t, I)} - \frac{1}{\mu_B(I)} \right) \right.$$

$$\left. + \left( \frac{Y_A(t, I)^2}{\lfloor It \rfloor^2} - \mu_A(I)^2 \right) \frac{1}{\mu_B(I)} \right|$$

$$\le s \left( s + \mu_A(I) + \frac{1}{\mu_B(I)} \right),$$

which is arbitrarily small as $\mu_A(I), \mu_B(I) \to \mu_A, \mu_B < \infty$. Therefore, almost surely,

$$\lim_{I \to \infty} \frac{q(\mathcal{S}_{\lfloor It \rfloor})}{I \mu_A(I)^2 / \mu_B(I)} = t.$$

The proof concludes noticing that

$$q(\mathcal{I}) = \frac{I \mu_A(I)^2}{\mu_B(I)},$$

and that $\mathbf{K} = \lfloor IU \rfloor$ with $U \sim \mathrm{U}([0, 1])$. $\qquad\square$

## C.3 Proof of Theorem 5

To prove Theorem 5, we need two preliminary results: Lemma 3, which itself needs two supplementary results (Lemmas 1 and 2), and Lemma 4, which is directly proved.

### C.3.1 Technical lemmata

**Lemma 1.** *Consider a set of $I$ values $N = \{n_1, \dots, n_I\}$. Let $X_1, \dots, X_k$ and $Y_1, \dots, Y_k$ denote, respectively, $k$ random samples with and without replacement from $N$. For any continuous and convex function $f$, it follows,*

$$\mathbb{E}\left[f\left(\sum_{i=1}^{k} Y_i\right)\right] \leq \mathbb{E}\left[f\left(\sum_{i=1}^{k} X_i\right)\right]$$

*Proof.* The proof follows from [14]. $\qquad\square$

**Lemma 2.** *Let $I \in \mathbb{N}$, $N := \{n_1, \dots, n_I\} \in \mathbb{R}_+^I$, $\mu = \frac{1}{I}\sum_{i=1}^{I} n_i$ be their mean value and $\sigma^2 = \frac{1}{I}\sum_{i=1}^{I}(n_i - \mu)^2$ be their variance. For $k \in \{0, \dots, n\}$, let $\mathcal{S}_k \sim \mathrm{U}(\{S_k \subseteq [I] : |S_k| = k\})$ be a uniform random variable on the subsets of $\{1, \dots, I\}$ of size $k$, and $n_{\mathcal{S}_k} = \sum_{i \in \mathcal{S}_k} n_i$ be the random variable defined by the sum of the elements of $\mathcal{S}_k$. Let $\mathbf{K} \sim \mathrm{U}(\{0, \dots, I\})$ and define $\mathbf{Y} = n_{\mathcal{S}_{\mathbf{K}}}$. Then,*

$$\mathbb{E}[\mathbf{Y} - \mu\mathbf{K} \mid \mathbf{K} = k] = 0, \tag{12}$$

$$\mathbb{E}\left[(\mathbf{Y} - \mu\mathbf{K})^2 \mid \mathbf{K} = k\right] \leq k\sigma^2. \tag{13}$$

*Proof.* We prove (12) directly.

$$\begin{aligned}
\mathbb{E}[\mathbf{Y} \mid \mathbf{K} = k] &= \sum_{S_k \subseteq [I]:|S_k|=k} n_{S_k} \frac{1}{\binom{I}{k}} = \frac{1}{\binom{I}{k}} \sum_{S_k \subseteq [I]:|S_k|=k} \sum_{i \in S_k} n_i \\
&= \frac{1}{\binom{I}{k}} \sum_{i \in [I]} \sum_{\substack{S_k \subseteq [I]:|S_k|=k \\ i \in S_k}} n_i \\
&= \frac{1}{\binom{I}{k}} \sum_{i \in [I]} n_i \binom{I-1}{k-1} \\
&= \frac{(I-k)!k!}{I!} \cdot \frac{(I-1)!}{(k-1)!(I-k)!} \sum_{i \in [I]} n_i = \mu k.
\end{aligned}$$

Thus, (12) follows as $\mathbb{E}[\mu\mathbf{K} \mid \mathbf{K} = k] = \mu k$. To prove (13), let $(\mathbf{X}_i)_{i=1}^k$ be $k$ independent samples from the set $N$. From Lemma 1 it holds,

$$\mathbb{E}\left[(\mathbf{Y} - \mu\mathbf{K})^2 \mid \mathbf{K} = k\right] \leq \mathbb{E}\left[\left(\mu\mathbf{K} - \sum_{i=1}^{\mathbf{K}} \mathbf{X}_i\right)^2 \mid \mathbf{K} = k\right] = \mathbb{E}\left[\left(\sum_{i=1}^{\mathbf{K}} (\mu - \mathbf{X}_i)\right)^2 \mid \mathbf{K} = k\right].$$

Therefore,

$$\begin{aligned}
\mathbb{E}\left[(\mathbf{Y}-\mu\mathbf{K})^2 \mid \mathbf{K} = k\right] &\leq \mathbb{E}\left[\left(\sum_{i=1}^{\mathbf{K}}\sum_{j=1}^{\mathbf{K}} (\mu - \mathbf{X}_i)(\mu - \mathbf{X}_j)\right) \mid \mathbf{K} = k\right] \\
&= \mathbb{E}\left[\left(\sum_{i=1}^{\mathbf{K}}\sum_{j=1}^{\mathbf{K}} \left(\mu^2 - \mu(\mathbf{X}_i + \mathbf{X}_j) + \mathbf{X}_i\mathbf{X}_j\right)\right) \mid \mathbf{K} = k\right] \\
&= \sum_{i=1}^{k}\sum_{j=1}^{k} \left(\mu^2 - \mu(\mathbb{E}[\mathbf{X}_i \mid \mathbf{K} = k] + \mathbb{E}[\mathbf{X}_j \mid \mathbf{K} = k]) + \mathbb{E}[\mathbf{X}_i\mathbf{X}_j \mid \mathbf{K} = k]\right) \\
&= \sum_{i=1}^{k}\sum_{j=1}^{k} \left(\mu^2 - \mu(\mathbb{E}[\mathbf{X}_i] + \mathbb{E}[\mathbf{X}_j]) + \mathbb{E}[\mathbf{X}_i\mathbf{X}_j]\right) \\
&= \sum_{i=1}^{k} \left(\mu^2 - 2\mu\mathbb{E}[\mathbf{X}_i] + \mathbb{E}[\mathbf{X}_i^2]\right) + \sum_{i=1}^{k}\sum_{\substack{j=1 \\ j \neq i}}^{k} \left(\mu^2 - \mu(\mathbb{E}[\mathbf{X}_i] + \mathbb{E}[\mathbf{X}_j]) + \mathbb{E}[\mathbf{X}_i]\mathbb{E}[\mathbf{X}_j]\right)
\end{aligned}$$

$$= \sum_{i=1}^{k} \mathbb{E}\left[(\mu - \mathbf{X}_i)^2\right] + \sum_{i=1}^{k}\sum_{\substack{j=1\\j\neq i}}^{k} \left(\mu^2 - 2\mu^2 + \mu^2\right)$$

$$= \sum_{i=1}^{k} \mathbb{E}\left[(\mu - \mathbf{X}_i)^2\right] = \sum_{i=1}^{k} \text{Var}\left(\mu - \mathbf{X}_i\right) = k\sigma^2.$$

The steps come from rearranging the terms, using the independence of $\mathbf{X}_i$ with respect to $\mathbf{K}$, the independence of $\mathbf{X}_i, \mathbf{X}_j$ for $i \neq j$, and finally that $\mathbb{E}[\mathbf{X}_i] = \mu$ and $\text{Var}(\mathbf{X}_i) = \sigma^2$. $\qquad\square$

**Lemma 3.** *Let $I \in \mathbb{N}$, $N := \{n_1, \dots, n_I\} \in \mathbb{R}_+^I$, and define,*

$$\mu = \frac{1}{I}\sum_{i=1}^{I} n_i, \quad \sigma^2 = \frac{1}{I}\sum_{i=1}^{I}(n_i - \mu)^2, \quad n^{max} = \max_{i\in\mathcal{I}} n_i,$$

$$R := \max_{i\in[I]} |n_i - \mu|, \quad \tau = \max_{i\in[I]} n_i / \min_{i\in[I]} n_i.$$

*Consider $\mathcal{S}_{\mathbf{K}}$, $n_{\mathcal{S}_{\mathbf{K}}}$, $\mathbf{K}$, and $\mathbf{Y}$ as in Lemma 2. Let $w : \mathbb{R}_+ \to \mathbb{R}$ be a function in $\mathcal{C}^2$, increasing, and suppose there exists $\kappa \in \mathbb{R}_+$, such that,*

$$\left|w^{(2)}(n)\right| \leq \frac{\kappa}{n^2}, \forall n > 0,$$

*where $w^{(k)}$ is the k-th derivative of $w$. Then, it holds,*

$$\left|\mathbb{E}[w(\mu\mathbf{K}) - w(\mathbf{Y})]\right| \leq \frac{\kappa}{2\mu^2 I}\left(9\sigma^2(1 + \ln(I)) + \frac{2R^2\tau^2}{n^{max}}\right).$$

*Proof.* The proof considers a second-order Taylor extension of $w$ at $\mu k$ to recover the expected value of $\mathbb{E}[w(\mu\mathbf{K}) - w(\mathbf{Y})]$. Noticing that the first derivative has a null expected value, the upper bound stated on the Lemma comes from bounding the expected value of the second derivative.

The Taylor-Lagrange Theorem on $w$ at $\mu k > 0$ provides,

$$w(y) = w(\mu k) + w^{(1)}(\mu k)(\mu k - y) + w^{(2)}(\tau)\frac{(\mu k - y)^2}{2},$$

for some $\tau$ between $y$ and $\mu k$. Therefore, there exists a random variable T, almost surely between $\mu\mathbf{K}_+$ and $\mathbf{Y}$, such that,

$$\mathbb{E}[w(\mathbf{Y}) - w(\mu\mathbf{K}_+)] = \mathbb{E}\left[w^{(1)}(\mu\mathbf{K}_+)(\mu\mathbf{K}_+ - \mathbf{Y}) + \frac{1}{2}w^{(2)}(\text{T})(\mu\mathbf{K}_+ - \mathbf{Y})^2\right],$$

where $\mathbf{K}_+$ corresponds to $\mathbf{K}$ conditioned to be positive. To avoid overcharging the notation, we drop the index from $\mathbf{K}_+$. We observe that,

$$\mathbb{E}\left[w^{(1)}(\mu\mathbf{K})(\mu\mathbf{K} - \mathbf{Y})\right] = \mathbb{E}\left[\mathbb{E}\left[w^{(1)}(\mu\mathbf{K})(\mu\mathbf{K} - \mathbf{Y}) \mid \mathbf{K} = k\right]\right]$$

$$= \mathbb{E}\left[w^{(1)}(\mu k)\mathbb{E}\left[(\mu\mathbf{K} - \mathbf{Y}) \mid \mathbf{K} = k\right]\right] = 0,$$

by Lemma 2, Equation (12). Therefore,

$$\left|\mathbb{E}[w(\mathbf{Y}) - w(\mu\mathbf{K})]\right| = \frac{1}{2}\left|\mathbb{E}\left[w^{(2)}(\text{T})(\mu\mathbf{K} - \mathbf{Y})^2\right]\right|$$

$$\leq \frac{1}{2}\mathbb{E}\left[\left|w^{(2)}(\text{T})\right|(\mu\mathbf{K} - \mathbf{Y})^2\right]$$

$$\leq \frac{1}{2}\mathbb{E}\left[\frac{\kappa}{\text{T}^2}(\mu\mathbf{K} - \mathbf{Y})^2\right] = \frac{\kappa}{2}\mathbb{E}\left[\frac{1}{\text{T}^2}(\mu\mathbf{K} - \mathbf{Y})^2\right].$$

Setting $\mathbf{I} := \left\{|\mu\mathbf{K} - \mathbf{Y}| \leq \frac{1}{2}(\mu\mathbf{K} + \mathbf{Y})\right\}$, the previous expected value can be expressed as,

$$\mathbb{E}\left[\frac{1}{\text{T}^2}(\mu\mathbf{K} - \mathbf{Y})^2\right] = \mathbb{E}\left[\frac{1}{\text{T}^2}(\mu\mathbf{K} - \mathbf{Y})^2 \cdot \mathbf{I}\right] + \mathbb{E}\left[\frac{1}{\text{T}^2}(\mu\mathbf{K} - \mathbf{Y})^2 \cdot \mathbf{I}^c\right].$$

We deal with each term separately. Notice that, as T is almost surely between $\mathbf{Y}$ and $\mu\mathbf{K}$,

$$|\mu\mathbf{K} - \mathbf{Y}| \leq \frac{1}{2}(\mu\mathbf{K} + \mathbf{Y}) \implies \mathrm{T} \geq \frac{1}{3}\mu\mathbf{K}.$$

Thus,

$$\mathbb{E}\left[\frac{(\mu\mathbf{K} - \mathbf{Y})^2}{\mathrm{T}^2} \cdot \mathbf{I}\right] \leq \mathbb{E}\left[\frac{(\mu\mathbf{K} - \mathbf{Y})^2}{(\frac{\mu\mathbf{K}}{3})^2} \cdot \mathbf{I}\right] = \frac{9}{\mu^2}\sum_{k=1}^{I}\frac{1}{I} \cdot \mathbb{E}\left[\frac{(\mu k - \mathbf{Y})^2}{k^2} \cdot \mathbf{I} \mid \mathbf{K} = k\right]$$

$$\leq \frac{9}{I\mu^2}\sum_{k=1}^{I}\mathbb{E}\left[\frac{(\mu k - \mathbf{Y})^2}{k^2} \mid \mathbf{K} = k\right]$$

$$\leq \frac{9}{I\mu^2}\sum_{k=1}^{I}\frac{k\sigma^2}{k^2}$$

$$\leq \frac{9\sigma^2}{I\mu^2} \cdot (1 + \ln(I)).$$

Regarding the second term, denote $\overline{n} := \max_{i \in \mathcal{I}} n_i$ and $\underline{n} := \min_{i \in \mathcal{I}} n_i$. As $\mathbf{K}\underline{n} \leq \min\{\mu\mathbf{K}, \mathbf{Y}\} \leq$ T, we have,

$$\mathbb{E}\left[\frac{(\mu\mathbf{K} - \mathbf{Y})^2}{\mathrm{T}^2} \cdot \mathbf{I}^c\right] \leq \mathbb{E}\left[\frac{(R\mathbf{K})^2}{\mathrm{T}^2} \cdot \mathbf{I}^c\right] \leq \mathbb{E}\left[\frac{(R\mathbf{K})^2}{(\underline{n}\mathbf{K})^2} \cdot \mathbf{I}^c\right]$$

$$= \frac{R^2}{I\underline{n}^2}\sum_{k=1}^{I}\mathbb{E}\left[\frac{1}{k^2}k^2 \cdot \mathbf{I}^c \mid \mathbf{K} = k\right]$$

$$= \frac{R^2}{I\underline{n}^2}\sum_{k=1}^{I}\mathbb{P}\left(|\mu k - \mathbf{Y}| > \frac{1}{2}(\mu k + \mathbf{Y}) \mid \mathbf{K} = k\right)$$

$$\leq \frac{R^2}{I\underline{n}^2}\sum_{k=1}^{I}\mathbb{P}\left(|\mu k - \mathbf{Y}| > \frac{\mu k}{2} \mid \mathbf{K} = k\right)$$

$$\leq \frac{R^2}{I\underline{n}^2}\sum_{k=1}^{I}\exp\left(-\frac{\mu^2 k}{2\overline{n}}\right)$$

$$= \frac{2R^2\tau^2}{I\mu^2\overline{n}}\sum_{k=1}^{I}\frac{\mu^2}{2\overline{n}}\exp\left(-\frac{\mu^2 k}{2\overline{n}}\right)$$

$$\leq \frac{2R^2\tau^2}{I\mu^2\overline{n}}\int_0^{\infty}\frac{\mu^2}{2\overline{n}}\exp\left(-\frac{\mu^2 k}{2\overline{n}}\right)dk = \frac{2R^2\tau^2}{I\mu^2\overline{n}},$$

as the integral corresponds to the cumulative distribution function of an exponential random variable of parameter $\lambda = \mu^2/2\overline{n}$. The upper bound on the theorem's statement is obtained when gathering all together. $\qquad\square$

**Lemma 4.** *Let* $w : \mathbb{R}_+ \to \mathbb{R}_+$ *be a smooth and increasing function such that*

$$\lim_{n \to \infty} n^2|w^{(2)}(n)| < \infty.$$

*Then, there exists* $\kappa > 0$ *such that* $n^2|w^{(2)}(n)| \leq \kappa$.

*Proof.* Notice that the assumptions imply, in particular, that $|w^{(2)}(n)|$ is bounded. We argue by contradiction. Suppose that for any $m > 0$, there exists $n_m$ such that

$$n_m^2|w^{(2)}(n_m)| > m.$$

Suppose the sequence $(n_m)_m$ converges to a point $n^*$. Then,

$$\lim_{m \to \infty} n_m^2|w^{(2)}(n_m)| > \lim_{m \to \infty} m = \infty,$$

which is a contradiction with $|w^{(2)}(n)|$ being bounded. Therefore, necessarily $(n_m)_m$ has to diverge. However, this implies,

$$\lim_{n\to\infty} n^2 |w^{(2)}(n)| = \lim_{m\to\infty} n_m^2 |w^{(2)}(n_m)| > \lim_{m\to\infty} m = \infty,$$

obtaining again a contradiction. $\qquad\square$

### C.3.2 Proof of Theorem 5

We are ready to prove Theorem 5.

**Theorem 5.** *Under Assumption H1, there exists a constant $\kappa > 0$, such that, for any $i \in \mathcal{I}$, it holds,*

$$\left| \varphi_i - \psi_i \right| \le \frac{\kappa}{(I-1)\mu_{-i}^2} \left( \sigma_{-i}^2 (1 + \ln(I-1)) + \zeta_{-i} \right),$$

*where $\varphi_i$ and $\psi_i$ are respectively the Shapley value and the* `DU-Shapley` *of player $i$, $\mu_{-i} = \frac{1}{I-1}\sum_{j\in\mathcal{I}\backslash\{i\}} n_j$ is the average dataset size of other players, $\sigma_{-i}^2 = \frac{1}{I-1}\sum_{j\in\mathcal{I}\backslash\{i\}}(n_j - \mu_{-i})^2$ its empirical variance, and $\zeta_{-i}$ measures the variability of the dataset sizes across players. Formally, it is defined as*

$$\zeta_{-i} := R_{-i}^2 \frac{\tau_{-i}^2}{4n_{-i}^{\max}}$$

*where $R_{-i} := \max_{j\in\mathcal{I}\backslash\{i\}} |n_j - \mu_{-i}|$, $n_{-i}^{\max} := \max_{j\in\mathcal{I}\backslash\{i\}} n_j$, and $\tau_{-i} := \frac{n_{-i}^{\max}}{\min_{j\in\mathcal{I}\backslash\{i\}} n_j}$.*

*Proof.* Under Assumption **H**1, Lemma 4 implies the existence of $\kappa > 0$ such that the value function $w$ satisfies all assumptions from Lemma 3. Theorem 5 comes from (a) noticing that

$$\varphi_i = \mathbb{E}[w(\mathbf{Y}_{-i} + n_i) - w(\mathbf{Y}_{-i})], \quad \psi_i = \mathbb{E}[w(\mathbf{K}\mu_{-i} + n_i) - w(\mathbf{K}\mu_{-i})],$$

where $\mathbf{K} \sim \mathrm{U}([I-1])$ and $\mathbf{Y}_{-i} = n_{\mathcal{S}_{\mathbf{K}}^{(i)}}$ with $\mathcal{S}_{\mathbf{K}}^{(i)}$ taking values on the subsets of $\mathcal{I} \backslash \{i\}$ of size $\mathbf{K}$, (b) writing

$$|\varphi_i - \psi_i| \le |\mathbb{E}[w(\mathbf{Y} + n_i) - w(\mathbf{K}\mu_{-i} + n_i)]| + |\mathbb{E}[w(\mathbf{Y}) - w(\mathbf{K}\mu_{-i})]|,$$

and (c) applying Lemma 3 to each of the expected values, as the function $n \to w(n + n_i)$ also satisfies **H**1. $\qquad\square$

