# OpenReview forum: "DU-Shapley: A Shapley Value Proxy for Efficient Dataset Valuation"
_NeurIPS.cc/2024/Conference — NeurIPS 2024 poster_

### Official Review · Reviewer_Q5wB · 2024-06-17

**Soundness:** 2
**Presentation:** 3
**Contribution:** 3
**Rating:** 5
**Confidence:** 3

**Summary:**

This paper studies the problem of dataset valuation, a variant of data valuation, where the contributions of multiple datasets, instead of single data points, to a metric such as performance of a machine learning prediction model are quantified. The authors analyze the computation of the Shapley value, where each player represents a single dataset. In this context, it is argued that the utility function (or game) is essentially determined by the size of each dataset, which is theoretically supported by three general synthetic use-cases involving piece-wise constant functions, homogeneous data distributions, and linear regression with noisy labels, related to local differential privacy. The author present "Discrete Uniform" Shapley values (DU-Shapley), which approximates the Shapley value under this assumption efficiently. If the assumption on the utility function is satisfied, then the method can be further simplified by constructing suitable intervals instead of sampling, where the authors show for all use-cases asymptotic convergence (for infinite number of datasets) and non-asymptotic theoretical guarantees for the first and second use-case. The authors then evaluate their approach empirically, on multiple benchmark datasets against permutation-based Monte Carlo approaches, and on OpenDataVal benchmark tasks.

**Strengths:**

- Dataset valuation is an important variant of data valuation and analyzing the structural properties of the involved utility function is an important research question.
- The paper is well-structured and the mathematical notation is precise and understandable
- The paper presents interesting theoretical results for synthetic use-cases, which have direct consequences for the computatability
- The empirical performance on OpenDataVal tasks is promising

**Weaknesses:**

- The baselines in the experimental evaluation of the Shapley value approximation are limited. The authors only consider permutation sampling (with antithetic sampling as a variant) as the main competitor. However, there exist a variety of other methods, cf. evaluations in [1], that have been applied in data valuation and outperform this approach
- There is no evaluation, if the assumptions / structures of DU-Shapley specific to dataset valuation are apparent in real-world applications. Especially, it remains unclear if the synthetic use-cases are related to real-world applications.
- Section 3.2 is hard to follow, especially the derivation of (10) from Definition 3. In particular, it remains unclear to me, how Definition 3 and (10) are the same estimator, and what is the intuition behind Definition 3, which is used in real-world applications.

**Minor Comments**

- In Table 1, it is unclear which values are bold (top-2 or best)? It seems inconsistent.
- Related work could be improved, there are several algorithms for Shapley values, cf. the methods in [1,2,3].
- Instead of saying "exponentially reduce the complexity", e.g. line 62,  it would be better to state the complexity


[1] Li, W., & Yu, Y. (2023, October). Faster Approximation of Probabilistic and Distributional Values via Least Squares. In The Twelfth International Conference on Learning Representations.

[2] Chen, H., Covert, I. C., Lundberg, S. M., & Lee, S. I. (2023). Algorithms to estimate Shapley value feature attributions. Nature Machine Intelligence, 5(6), 590-601.

[3] Kolpaczki, P., Bengs, V., Muschalik, M., & Hüllermeier, E. (2024, March). Approximating the shapley value without marginal contributions. In Proceedings of the AAAI Conference on Artificial Intelligence (Vol. 38, No. 12, pp. 13246-13255).

**Questions:**

- Could you explain a bit more the intuition behind the DU-Shapley approach and Definition 3? In particular, the definition of $D^{(k)}$ is unclear to me, "is a set of data points uniformly sampled without replacement", it seems to me that it is the union of $k$ datasets rather than $k$ data points? If not, then $u$ is not defined for this setting? Maybe a pseudo-code algorithm would help.
- Could you elaborate on section 3.2? I did not understand the derivation of (10) from Definition 3 using the assumption. In particular, how does (10) not depend on the sampled data points anymore from Definition 3? These seem to be two different estimators.
- It seems that DU-Shapley has to be computed for every individual dataset separately? Could you comment on this limitation, and is there a way to circumvent it? I think this should be included in the limitations sections. Furthermore, the method then has quadratic complexity?

**Limitations:**

The author discuss limitations of their work, which centers around the applicability of the approach when dataset sizes and variances are strongly heterogeneous.

---

> ### Author Rebuttal · Authors · 2024-08-06
>
> We thank the anonymous reviewer for the remarks and questions. They are all addressed in detail in the following.
>
> 1. **The baselines in the experimental evaluation of the Shapley value approximation are limited**. Please remark in Table 2 that we have also considered KNN-Shapley and LOO. In addition, the experiments showed in the general answer consider the SVARM method 1], as suggered by reviewer VRG1.
>
> 2. **Evaluation of the assumptions in real-world applications**. Assumptions in the article have been done specifically to being able to deduce theoretical guarantees. However, our method is not limited to these assumptions and can perform well in real-world applications as our experiments show.
>
> 3. Concerning your **1st Question**. Definition 3 consists in (a) pooling all datasets $(D_j, j\neq i)$ together to create $D_{-i}$, (b) sampling $I$ datasets from $D_{-i}$, respectively, of sizes $|D^{(k)}| = k \cdot \mu_{-i}$ for $k \in (0,...,I-1)$, where $\mu_{-i} = \frac{1}{I-1}|D_{-i}|$, and (c) computing the average marginal contribution
> $$\frac{1}{I}\sum_{k=0}^I u(D^{(k)} \cup D_i) - u(D^{(k)}).$$
> In particular, each $D^{(k)}$ is obtained by sampling $k\cdot\mu_{-i}$ data points from $D_{-i}$ without replacement. Remark, therefore, that $D^{(k)}$ is not the union of $k$ datasets but a set of $k\cdot\mu_{-i}$ data points where each data point can come from any dataset $D_{j}$ for $j \in \mathcal{I}\setminus\{i\}$. Moreover, data points may belong to several sets $D^{(k)}$.
> We agree with the reviewer that a pseudo-code may help to understand DU-Shapley. Unfortunately, due to the space constraint, we have preferred to express DU-Shapley as a formula to being able to state our posterior theoretical results.
>
> 4. Concerning your **2nd Question**. We thank the reviewer for this question, as it has helped us to remark that we have left an old version of Equation (10) in the article. The updated version is,
> \begin{align*}
> \psi_i(w) = \frac{1}{I}\sum_{k=0}^{I-1} w(q^{(k)}) - w(q_{-i}^{(k)}) \text{ where } q^{(k)} := \frac{(\gamma_i n_i + \frac{k}{I-1}\sum_{j\in \mathcal{I}\setminus\{i\}} \gamma_j n_j)^2}{\gamma_i^2 n_i + \frac{k}{I-1}\sum_{j\in \mathcal{I}\setminus\{i\}} \gamma_j^2 n_j} \text{ and } q_{-i}^{(k)} := \frac{k}{I-1}\cdot q(\mathcal{I}\setminus\{i\}).
> \end{align*}
> The purpose of Equation (10) was to give an easier way to compute DU-Shapley. Remark the updated version coincides exactly with Definition 3 in the first two use-cases, i.e., when $\gamma_j = \gamma$ for all $j \in \mathcal{I}$. Indeed, as the random datasets $D^{(k)}$ have a fixed size and the value function only looks at the number of data points within the coalition, that is, $u(\mathcal{S}) = w(\sum_{i\in\mathcal{S}} n_i)$, we obtain,
> \begin{align*}
>     \psi_i(u) &= \frac{1}{I}\sum_{k=0}^{I-1} u(D^{(k)} \cup D_i) -  u(D^{(k)}) =\frac{1}{I}\sum_{k=0}^{I-1} w(|D^{(k)} \cup D_i|) -  w(|D^{(k)}|) = \frac{1}{I}\sum_{k=0}^{I-1} w(k\mu_{-i} + n_i) -  w(k\mu_{-i}) = \psi_i(w).
> \end{align*}
> For the third use-case, the updated version of Equation (10) comes from assuming that, for any $j \in \mathcal{I}$, $|D_j \cap D^{(k)}| = k \cdot \frac{n_j}{I-1}$, which holds with high probability for large values of $I$. We ensure none of the theoretical or numerical results is impacted by this typo as all of them are based on the general version given in Definition 3.
>
> 5. Concerning your **last question**, please refer to the general answer.
>
>
> [1] Kolpaczki, P., Bengs, V., Muschalik, M., & Hüllermeier, E. (2024). Approximating the Shapley Value without Marginal Contributions. Proceedings of the AAAI Conference on Artificial Intelligence, 38(12), 13246-13255.

---

> > ### Comment · Reviewer_Q5wB · 2024-08-12
> > **Clarification on 3.**
> >
> > I thank the authors for their response.
> >
> > **3.** Do I understand correctly that the DU-Shapley values are then highly probabilistic, since they strongly depend on the given sampled data points? In your theoretical analysis, you are taking the limit with respect to the number of players $I \to \infty$. Are you dealing with this randomness, or is it mitigated by the chosen structure of the value function (since it does not matter which data points are sampled)? Could you comment then, if this assumption is reasonable? I would expect that there is heterogeneity in the value of data points.

---

> > > ### Author Response · Authors · 2024-08-12
> > > **Follow-up answer on 3.**
> > >
> > > We thank the anonymous reviewer for the follow-up question.
> > >
> > > It is true that DU-Shapley is stochastic in the general case, so do Monte Carlo estimators of the Shapley value. Note that if the variance of DU-Shapley is too high, one could take an estimator of the mean of the DU-Shapley instead.
> > >
> > > In our non-asymptotic theoretical results, note that we are not using the general definition of DU-Shapley (Definition 3) but a specific, albeit general enough, case where $u(S) = w(q(S))$. In this scenario, the sampling procedure and its stochasticity is significantly improved via a proper discretisation. As such, as you correctly pointed out, this stochasticity is mitigated by the chosen structure of the value function.
> > > We believe that this assumption is quite reasonable as it applies to several ML use-cases as depicted in Sections 2.1.1 to 2.1.3.
> > >
> > > In the general scenario where only Definition 3 applies, we don't think that the sampling procedure and hence the stochasticity is a problem. In most use-cases, the players generate their data with their processes. Their values are determined by the quality of the generating processes, such as the precision and scope of the reporting, and the nature of the data. For example, the hospital sharing the data has a unit for patients with rare diseases.
> > > While there may be cases where the values are mostly brought by a few outliers, this is not the situation we were trying to solve with DU-Shapley.

---

> > > > ### Comment · Reviewer_Q5wB · 2024-08-12
> > > > **follow-up**
> > > >
> > > > I thank the authors again for the quick response. Is there evidence that this stochasticity is not a problem? If it does not make a difference, I would suppose that it is required that the value of each data point in the dataset is homogeneous. Wouldn't this undermine the general purpose of data valuation? In practice, I would expect that given a certain prediction task, some data is much more important than other. I am still struggling to accept that the value in real-world settings is mostly determined by the number of data points in each datasets.

---

> > > > > ### Author Response · Authors · 2024-08-12
> > > > > **Follow-up**
> > > > >
> > > > > We thank again the anonymous reviewer for the quick follow-up.
> > > > >
> > > > > *Is there evidence that this stochasticity is not a problem?*
> > > > >
> > > > > As correctly pointed out by the reviewer, the datasets (hence their value) owned by each player are not equivalent in the heterogeneous scenario which indeed pose some question on the uniform sampling procedure defined in Definition 3. In this paper, we provide empirical evidence that this stochasticity is not a problem through several experiments as recalled hereafter.
> > > > >
> > > > > In Section 4.1, we considered real-world datasets which have been split among players so that each player owned a heterogeneous partition of the global dataset. For instance, for classification tasks, we built those local datasets by considering label shift, i.e. p(x|y) fixed while p(y) is changing across players. For regression tasks, we considered feature shift, i.e. each player owns different local feature distributions p(x).
> > > > > As depicted by the empirical evidences in Table 1, this stochasticity does not pose any problem since DU-Shapley is able to provide relevant Shapley value approximations for each player.
> > > > >
> > > > > The same type of data heterogeneity has been considered in Section 4.2 with a larger number of players (I =100) and DU-Shapley still provides meaningful data values compared to existing dataset valuation baselines.

---

> > > > > > ### Comment · Reviewer_Q5wB · 2024-08-13
> > > > > > **Thank you!**
> > > > > >
> > > > > > Thank you for further clarification and addressing my concerns, I have increased my score by point. However, I still think the empirical analysis of these properties and approximation, as well as the baseline approximations, could be improved.

---

> ### Author Response · Authors · 2024-08-13
> **Thank you**
>
> We would like to thank the anonymous reviewer for this interesting discussion and for taking into account our answers to increase his/her score. As suggested by the reviewer, we will add complementary empirical evidences in the final version of the manuscript, if accepted.

---

### Official Review · Reviewer_VRG1 · 2024-07-05

**Soundness:** 3
**Presentation:** 3
**Contribution:** 3
**Rating:** 7
**Confidence:** 4

**Summary:**

The paper proposes a novel proxy for the Shapley value (SV) which is called discrete uniform (DU) Shapley. Unlike the SV, this proxy does not suffer from the exponential complexity problem of the SV. Of course the DU Shapley, therein does not capture all information entailed in the SVs for a cooperative game. The DU Shapley are motivated and derived from the setting of dataset valuation. Dataset valuation is a similar setting to data valuation, with the difference being that not individual data points are making up the player set but collection of data points. The work studies this dataset valuation setting and the DU Shapley theoretically and performs empirical experiments illustrating the effectiveness.

**Strengths:**

- **Theoretical Results:** The work studies the setting of dataset valuation in a theoretical and well-substantiated manner. While I think details of 2.1.1, 2.1.2, and 2.1.3 may be deferred to the appendix to expand on 2.2 and 2.3, the theoretical analysis of the dataset valuation problem is a good contribution. Further, the theoretical results regarding DU Shapley and the mechanisms behind it (for Example Figure 1 and Figure 2) support the work well.
- **Empirical Evaluation For Data Valuation:** While DU Shapley is not proposed for the data valuation task but rather dataset valuation, the work also contains evaluations with OpenDataVal highlighting that DU Shapley can also be applied to this setting. This is particularly interesting since these values are quite cheap to compute offering a good baseline for future work or an initial insight before applying more costly approaches.
- **Empirical Evaluation in real-world scenarios:** The paper shows how DU Shapley seems to be a viable approach to use as a proxy stand-in compared to other sampling-based baseline methods in traditional machine learning settings. This shows that even when the assumptions made on the value function are not fulfilled the estimates are still better than estimates from alternative methods. I still think better comparisons could have been made (c.f. Weakness Section).
- **Reproducible:** Working on a related problem with the same problem formulation, I observe the same results this paper hypothesizes and describes in its formulation. With increasing dataset size the worth of a dataset increases (though not monotonically but still increases). The proxy approach of DU Shapley seems to work very well in this setting. I suspect this also to hold for different data modalities like text or images, which could be a fruitful future research direction.

**Weaknesses:**

## Major Weaknesses
- **Weak Baselines** The empirical evaluation (comparison against sampling) compares the effectiveness of DU Shapley in with subpar and not interesting baselines. The evaluation does not include the KernelSHAP [1] method for computing SVs. KernelSHAP in many areas outperforms the sampling based estimators by a large margin. Further, as the work hypothesizes and arguments strongly that the size of a dataset (and thus also the coalition size) is important to determine the worth of it, methods like SVARM [2] that are derived for this setting are also missing from the empirical comparison. Both of these methods are not compared, which might strongly alleviate the performance increase of DU Shapley compared to the current baselines.
- **Weak Sampling Budget** It seems like the empirical evaluation comparing DU Shapley with the sampling-based black-box estimators (Section 4.1) uses a very low computational budget for the estimators. While it is true that when DU Shapley requires a very low budget, a fair comparison would want to compare the current state of the art in the same settings. However, DU Shapley is a proxy for the SV. There it is definitely interesting to also see when the state-of-the-art catches up with the proxy. While it is very obvious that the proxy DU Shapley value is much better in these very low dataset valuation settings. However, the other methods estimate the SV and might get better in a reasonable time (in terms of computational budget). This should be evaluated well with proper error/evaluation curves for different estimation budgets.

## Smaller Weaknesses
- **Too little Focus on Data Valuation vs. Dataset Valuation:** I think this work may benefit from a better high-level motivation, and looses itself in too much details (2.1.1, 2.1.2, and 2.1.3 make up 1.5 pages). These motivating examples could be shorter and placed in the appendix to make more room for a broader theoretical background delineating this works contribution from the classical data valuation setting better than the rather short intermission with Section 2.3. I would also think that actually plotting/visualizing these Dataset Valuation attribution values for a ground truth setting would help. It is not tangible how these values actually look for real-world scenarios from Section 4.2.


## References
- [1] https://proceedings.neurips.cc/paper/2017/file/8a20a8621978632d76c43dfd28b67767-Paper.pdf (KernelSHAP in my opinion is better explained here: https://proceedings.mlr.press/v130/covert21a.html)
- [2] https://ojs.aaai.org/index.php/AAAI/article/view/29225

**Questions:**

1. How does DU Shapley compare against stronger approximations like Kernel-based estimators (aka. KernelSHAP) or stratified methods like SVARM?
2. I am a bit confused by your statement in line 305. Did you run the MC estimations on 10 or 20 subset evaluations as the computational budget? The axis description in Figure 3 also does not help.

## Suggestions
I have run a similar setup for the cal. housing experiment as you do in Section 4.1. I compute the ground-truth SVs for a 10 player dataset valuation setting. The SVs are as follows (empty-set is manually set to 0):
- $D_1$: 0.05790964954064733
- $D_2$: 0.06817158889611828
- $D_3$: 0.07737767986447398
- $D_4$: 0.08361640115201878
- $D_5$: 0.08403403608007484
- $D_6$: 0.08978074706995469
- $D_7$: 0.08771940889382171
- $D_8$: 0.08772011962276798
- $D_9$: 0.09842460151765661
- $D_{10}$: 0.11083578157985912

The dataset sizes are monotonically increasing from $1$ to $10$, which is inline with the papers claims in regards to the behavior of the SVs. The SVs also increasing (though not so monotonically but still increase) mainly depending on the size of the datasets. Note that for this a gradient boosted tree (as implemented in xgboost) is fitted without any HPO. I suggest to add something similar to this in an illustrative/abstract way to show how the value function behaves and make this clear to a potentially unfamiliar reader too motivate this solution better.

**Limitations:**

The work contains a separate limitations section.

---

> ### Author Rebuttal · Authors · 2024-08-06
>
> We thank the anonymous reviewer for the remarks, questions, and suggestion. They are all addressed in detail in the following.
>
> 1. **Weak Baselines**. We thank the reviewer for the references. We have conducted experiments concerning the theoretical probability at which SVARM can guarantee an error equal to DU-Shapley's bias. Please refer to the general answer for the results.
>
> 2. **Weak Sampling Budget**. We have performed extra experiments comparing our method with Monte Carlo and the improved Jia et al.'s method in [1] by looking at the number of iterations that both of them require to catch up DU-Shapley. Please refer to the general answer for the results.
>
> 3. **Too little Focus on Data Valuation vs. Dataset Valuation** We agree with the reviewer that the article may benefit from a longer discussion about data valuation vs dataset valuation. We will extend this section in the final version if the article is accepted.
>
> 4. Concerning your **1st Question**. Please refer to the general answer concerning SVARM. Unfortunately, due to the time of rebuttals, we have not been able to study KernelSHAP, but it will be considered in future versions of the article.
>
> 5. Concerning your **2nd Question**. Both. For $I = 10$ players we have done 10 subset evaluations while for $I = 20$ players, we have done $20$.
>
> [1] Wang, Jiachen T., and Ruoxi Jia. "A Note on" Towards Efficient Data Valuation Based on the Shapley Value''." arXiv preprint arXiv:2302.11431 (2023).

---

> > ### Comment · Reviewer_VRG1 · 2024-08-09
> > **Follow Up Question and Suggestion**
> >
> > Thank you for your reply!
> >
> > 1. Maybe, I am missing something simple here, but why are you measuring the time/iterations until the baselines reach **the bias of DU-Shapley** and not the approximation error against a/the ground truth in your _attached pdf/general statement_?
> >
> > 2. Thanks for the clear-up with the sampling budget. I have to say that this is extremely small in terms of number of Model Evaluations. For an improved version, I strongly suggest to reevaluate the approximation experiments with much higher budgets. Even if this is in some sense unfair for your method, because it would greatly improve the paper for practitioners that are currently approximating.

---

> > > ### Author Response · Authors · 2024-08-12
> > > **Follow Up**
> > >
> > > We thank you for your reply.
> > > 1. We agree with the reviewer that checking numerically the number of iterations that other methods require to achieve DU-Shapley's approximation error would be a great joint to the paper. Unfortunately, we have not had enough time to conduct this experiments during the rebuttals, but they will be added to the final version of the article. In exchange, we have prepared this preliminary *theoretical* experiment as we were sure to achieve it on time.
> > >
> > > 2. We believe there may have been a misunderstanding. For Table 1, as explained above, we have given each method a budget equal to the number of players. Therefore, for instances with 10 players, all methods had a budget equal to 10, while for those with 20 players, all methods had a budget equal to 20. We set this constraint to compare all methods with DU-Shapley at the same budget, as our method requires only $I$ function valuations. However, the experiment in Figure 3 presents the MSE achieved by each method at different budgets when approximating the Shapley value of one player. DU-Shapley was computed exactly, i.e., using $I$ function valuations everytime but MC-based methods have been computed at different budgets ranging from 10% of the budget of DU-Shapley up to 10 times the budget of DU-Shapley.

---

### Official Review · Reviewer_WTCU · 2024-07-17

**Soundness:** 4
**Presentation:** 4
**Contribution:** 3
**Rating:** 6
**Confidence:** 4

**Summary:**

The paper studies dataset evaluation with Shapley value. They assume that each dataset contains i.i.d. data points from a distribution and exploit the knowledge about the structure of the underlying data distribution to design more efficient Shapley value estimators. Based on asymptotic analysis of Shapley values of datasets, they propose an approximation of Shapley values of datasets. They prove estimation error bounds for non-asymptotic settings. The estimation accuracy is tested via numerical experiments in certain settings in comparison to other approaches. They also test the effectiveness of the approach for noisy label detection, dataset removal, and dataset addition.

**Strengths:**

The paper explores a novel setting in which Shapley values are computed for datasets comprising i.i.d. data points. Leveraging the distributional structures of datasets, they propose an approximation method that reduces running time. Their theoretical analysis is novel and strong, offering valuable insights into the distribution of Shapley values for datasets under some assumptions. They show by experimental results that this approximation method outperforms other MC-based approaches with computational power is limited.

**Weaknesses:**

In the context of dataset evaluation, previous data point Shapley approximation methods can still be applied by treating the dataset as a single entity. The main advantage of this new approach appears to be its improved computational efficiency compared to earlier Data Shapley methods. However, the computational efficiency improvement is not thoroughly discussed or demonstrated through experiments, which primarily focus on estimation accuracy.

The theoretical analysis relies on certain distributional assumptions, potentially limiting the approach's applicability. For instance, the accuracy of the estimation seems dependent on having a large number of data providers. In contrast, previous Data Shapley approximation methods, such as those proposed by Jia et al. (2019), do not appear to require these assumptions. Furthermore, the difference in the number of retraining iterations, O(n log n) versus O(n), is not substantial.

The use cases for dataset valuation are not well articulated. Common tasks like noisy label detection, data removal, and data addition are typically considered at the data point level. It is unclear why the data needs to be removed as a set rather than individually.

**Questions:**

1. What’s the reason for choosing 20 SGD steps and 20 boosting iterations? Will the experiment results significantly change when the numbers increase?
2. Could you explain the improvement in computational efficiency compared to Jia et al. 2019?

**Limitations:**

Yes.

---

> ### Author Rebuttal · Authors · 2024-08-06
>
> We thank the anonymous reviewer for the remarks and questions. They are all addressed in detail in the following.
>
> 1. Concerning the **computational efficiency improvements** of our method, we have conducted new experiments. Please refer to the general answer for the results.
>
> 2. **Distributional assumptions**. Again, we kindly ask to refer to the general answer. We remark that, although former methods can achieve a lower complexity when computing all Shapley values, our method achieves much more precise approximations.
>
> 3. **Dataset valuation use-cases**. Dataset addition and removal find several applications in industrial scenarios or theoretical domains such as federated learning and collaborative learning.
>
> 4. Answer to your **1st Question**. We have considered the same setting as in [1], with 20 SGD steps and 20 boosting iterations instead of 100 for speed of computations. Concerning the results sensibility to the parameters, the ranking between the methods does not depend on this choice but only the relative difference between them.
>
> 5. Answer to your **2nd Question**. Please refer to our general answer.
>
> [1] https://arxiv.org/pdf/2104.12199

---

> > ### Comment · Reviewer_WTCU · 2024-08-11
> >
> > Thank you for the rebuttal. The new experiments mostly addressed my concerns. I thereby updated my score.

---

### Official Review · Reviewer_C64x · 2024-07-25

**Soundness:** 3
**Presentation:** 2
**Contribution:** 3
**Rating:** 6
**Confidence:** 2

**Summary:**

This paper proposes a general dataset valuation method which only requires N number of utility function evaluations, where N is the number of players. The approach is motivated by a homogeneous setting for linear regression, but experiments also show the effectiveness of such an approach in general settings.

**Strengths:**

The paper is easy to read and has a very clear structure. The topic is very important as existing data valuation approaches mostly focus on data point-level valuation, while this paper focus on dataset-level valuation.

**Weaknesses:**

I have reviewed this paper at ICML. The paper has addressed most of the issues in this revision and has a huge improvement. My remaining concerns are the following:

**Computational complexity.**
While DU-Shapley only requires $I$ utility function evaluations for a single data owner, if we want to evaluate all data owners it still requires $I^2$ function evaluations. This is worse than the group-testing estimator introduced in [1] (a refined version can be found in [2]), and I believe [3] has already reduced the complexity to $O(I \log I)$.

I did not verify the correctness of the results in Section 3.3, but the expressions seem to be reasonable.

[1] Jia, Ruoxi, et al. "Towards efficient data valuation based on the shapley value." The 22nd International Conference on Artificial Intelligence and Statistics. PMLR, 2019.

[2] Wang, Jiachen T., and Ruoxi Jia. "A Note on" Towards Efficient Data Valuation Based on the Shapley Value''." arXiv preprint arXiv:2302.11431 (2023).

[3] Li, Weida, and Yaoliang Yu. "Faster Approximation of Probabilistic and Distributional Values via Least Squares." The Twelfth International Conference on Learning Representations.

**Questions:**

What's the specific MC estimator being used for computing Data Shapley in the experiment?

**Limitations:**

See weaknesses.

---

> ### Author Rebuttal · Authors · 2024-08-06
>
> We thank the anonymous reviewer for the remarks and questions.
>
> 1. Regarding the **computational complexity** discussion, please refer to the general answer.
>
> 2. Regaring your **question**, the MC estimator used to estimate the Shapley values is the one presented in Equation (6), also coined DataShapley in the general answer or *baseline* in [1].
>
> [1] Jia, Ruoxi, et al. "Towards efficient data valuation based on the shapley value." The 22nd International Conference on Artificial Intelligence and Statistics. PMLR, 2019.

---

### Author Rebuttal · Authors · 2024-08-06

We thank the anonymous reviewers for the useful reviews. As the question concerning the complexity of computing the Shapley value of all players arose several times, we address it as a general reply. We provide several answers.

a) Approximation schemes such as Monte Carlo (DataShapley), DU-Shapley, SVARM [2], or the improved version of Jia et al.'s algorithm [1], are all based on a simple principle: a trade-off between accuracy and complexity of the approximations. In particular, even though DataShapley and the work in [1] require $O(I\log(I))$ function valuations to compute all Shapley values while our method uses $O(I^2)$, the comparison between them must be carefully done as nothing guarantees that the different methods achieve the same accuracy level. We illustrate the advantages of our method through two experiments.

1. **Complexity comparison of DU-Shapley, DataShapley (Monte Carlo), and the improved Jia et al.'s algorithm**. We have looked at the number of iterations that DataShapley (Monte Carlo, also named the \textit{baseline} in [3]) and the method in [1] require to achieve DU-Shapley's accumulated bias (in norm-2), formally given by,
\begin{align*}
    \mathrm{DU bias}(I) := \frac{\kappa}{I-1}\sqrt{\sum_{i\in \mathcal{I}} \frac{(9\sigma_{-i}^2(1+\log(I-1)) + \zeta_{-i})^2}{(\mu_{-i})^4}},
\end{align*}
by replacing $\varepsilon = \mathrm{DU bias}(I)$, respectively, in the formula in Section 4.1 in [3] and Equation 5 in [1], for $\delta \in \{0.01,0.05,0.1\}$. Since DU-Shapley's bias depends on the value function choice (due to the constant $\kappa$), we have considered a utility function motivated from our third use-case under the homogeneity assumption $\sigma_i/\varepsilon_i = \sigma_j/\varepsilon_j$ for all $i,j \in \mathcal{I}$, given by,
\begin{align*}
\forall S \subseteq \mathcal{I}, u(S) = w(n_S) = 1 - \frac{10^{k(\mathcal{I})}}{10^{k(\mathcal{I})} + n_S},
\end{align*}
where $n_S$ is the size of the data set $D_S := \cup_{i\in S} D_i$ and $k(\mathcal{I}) := \lfloor \log(\sum_{i\in\mathcal{I}}n_{i}) \rfloor - 1$ is a normalization factor depending on the total number of data points. Finally, we have shifted the utility function so it takes only positive values and $u(\varnothing) = 0$. Since DUbias depends on the number of data points per player, for each considered value of $I$, we have simulated 20 different sets of players, each time by sampling $n_i \sim \mathrm{U}(\{1,...,n_{max}\})$ for each $i \in \mathcal{I}$ and $n_{max} \in \{10,50,100\}$. The results are illustrated in Figure 1 in the attached PDF. The rows correspond to each value of $\delta$ while the columns to each value of $n_{max}$. We observe that in all tested instances, both methods require more than $I^2$ iterations to achieve the same error than DU-Shapley.

2. **Complexity comparison DU-Shapley and SVARM.** We have looked at the probability at which SVARM (Theorem 4 [2]) can ensure, after $I^2$ iterations (without considering the warm up as part of the budget), an error equal to DU-Shapley's accumulated bias. We have considered the same value function than in the previous experiment with $n_{max} \in (2\cdot 10^3,3\cdot 10^3,5\cdot 10^3,10^4)$ and 100 simulations of sets of players at each time. Figure 2 in the attached PDF shows the results.

b) **Approximating DU-Shapley**. It should be noted that the approximation of the Shapley value given by DU-Shapley can be leveraged further by approximating DU-Shapley itself with a coarser discrete scheme than the one specified in our method. For example, we can reduce the computing burden  from $I$ to $\sqrt{I}$ in Definition 3 by computing $\sqrt{I}$ marginal contributions only with data sets $\mathrm{D}^{(k)}$, respectively, of size $$k\cdot \frac{1}{\sqrt{I-1}}\sum_{j\in \mathcal{I}\setminus\{i\}}|\mathrm{D}_j|.$$

c) **Practical use-cases for dataset valuation**. We also point out that there are use cases where one is interested in the contribution of one specific dataset (for example, the contribution of Wikipedia to an LLM).
In this case, the relevant complexity notion is the computing cost for one player, as opposed to the computing cost for all players.

[1] Wang, Jiachen T., and Ruoxi Jia. "A Note on" Towards Efficient Data Valuation Based on the Shapley Value''." arXiv preprint arXiv:2302.11431 (2023).

[2] Kolpaczki, P., Bengs, V., Muschalik, M., & Hüllermeier, E. (2024). Approximating the Shapley Value without Marginal Contributions. Proceedings of the AAAI Conference on Artificial Intelligence, 38(12), 13246-13255.

[3] Jia, Ruoxi, et al. "Towards efficient data valuation based on the shapley value." The 22nd International Conference on Artificial Intelligence and Statistics. PMLR, 2019.

---

### Decision · Program_Chairs · 2024-09-25

**Decision:**

Accept (poster)

**Comment:**

The review for the paper were unanimously positive, and the paper should be accepted as at NeurIPS